# Scaling Generative Verifiers for Natural Language Mathematical Proof Verification and Selection

**Sadegh Mahdavi** [1 2]  **Branislav Kisacanin** [1 3]  **Shubham Toshniwal** [1]  **Wei Du** [1]  **Ivan Moshkov** [1]
**George Armstrong** [1]  **Renjie Liao** [† 2]  **Christos Thrampoulidis** [† 2]  **Igor Gitman** [† 1]

## Abstract

Large language models have achieved remarkable success on final-answer mathematical problems, largely due to the ease of applying reinforcement learning with verifiable rewards. However, the reasoning underlying these solutions is often flawed. Advancing to rigorous proof-based mathematics requires reliable proof verification capabilities. We begin by analyzing multiple evaluation setups and show that focusing on a single benchmark can lead to brittle or misleading conclusions. To address this, we evaluate both proof-based and final-answer reasoning to obtain a more reliable measure of model performance. We then scale two major generative verification methods (GenSelect and LLM-as-a-Judge) to millions of tokens and identify their combination as the most effective framework for solution verification and selection. We further show that the choice of prompt for LLM-as-a-Judge significantly affects the model's performance, but reinforcement learning can reduce this sensitivity. However, despite improving proof-level metrics, reinforcement learning does not enhance final-answer precision, indicating that current models often reward stylistic or procedural correctness rather than mathematical validity. Our results establish practical guidelines for designing and evaluating scalable proof-verification and selection systems. Code is available at https://github.com/NVIDIA-NeMo/Skills/tree/main/recipes/proof-gen-verification

## 1. Introduction

Large Language Models (LLMs) have achieved remarkable progress in mathematical reasoning tasks, reaching near-saturation on high-school competition benchmarks such as MATH (Hendrycks et al., 2021) and AIME (OpenAI, 2025b). This success is driven in large part by the simplicity of evaluating final-answer problems: correctness can be checked automatically, either via string matching or with lightweight equivalence checks (Seed et al., 2025). This makes such tasks well-suited for reinforcement learning with verifiable rewards (RLVR) (Shao et al., 2024), enabling steady improvements in model performance.

Closer inspection, however, reveals a major gap: LLMs often arrive at correct answers through flawed reasoning (Guo et al., 2025b). Common failure modes include circular reasoning where models assume what they need to prove, pattern-matching by testing small cases rather than providing formal proofs, adding extra assumptions to the problem, and fabricating theorems or lemmas that sound plausible but are incorrect (Mahdavi et al., 2025). Furthermore, advancing to harder competitions such as the IMO or USAMO requires a shift in focus: solutions demand rigorous proofs, not just final answers. Unlike final-answer verification, which can be automated via simple checks, proof verification is more challenging as it requires assessing logical structure, mathematical rigor, and the validity of each reasoning step.

Developing reliable verification mechanisms in this context could enable RLVR-style approaches to enhance proof generation, mirroring their demonstrated success with final-answer tasks. This motivates the investigation of two critical verification settings: (1) *Selection*: Given multiple candidate proofs, identifying the most sound one, valuable for test-time compute scaling and curating high-quality data for self-improvement or expert iteration (Zelikman et al., 2022; Dong & Ma, 2025). (2) *Verification*: Given a single proof, determining its correctness, enabling precise reward signals for policy-gradient reinforcement learning approaches such as GRPO (Shao et al., 2024). Both settings have received substantial recent attention (Ma et al., 2025; Dekoninck et al., 2025), with LLM-based generative verification emerging as particularly promising (Zhang et al., 2024; Lifshitz

---

[1]NVIDIA  [2]University of British Columbia  [3]Institute for AI R&D of Serbia. Correspondence to: Igor Gitman <igitman@nvidia.com>, Christos Thrampoulidis <cthrampo@ece.ubc.ca>, Renjie Liao <renjie.liao@ubc.ca>.

*Proceedings of the 43rd International Conference on Machine Learning*, Seoul, South Korea. PMLR 306, 2026. Copyright 2026 by the author(s).

et al., 2025).

In this work, we systematically study LLM-based proof verification and selection. Our contributions are as follows:

- We establish a rigorous evaluation framework that exposes significant limitations in existing datasets, demonstrating the necessity of a multi-faceted evaluation suite to mitigate exploitable correlations and avoid drawing conclusions from insufficiently sized datasets.

- We analyze multiple variants of LLM-as-a-Judge methodologies for both proof selection and verification, revealing that prompt design exerts substantial influence on performance, and that reinforcement learning fine-tuning can effectively reduce this sensitivity.

- We investigate diverse strategies for scaling test-time compute in proof selection, demonstrating that a synergistic combination of LLM-as-a-Judge with GenSelect tournaments and parallel judgments achieves an improved compute trade-off.

## 2. Related Work

**Test-Time Scaling for Mathematical Reasoning.** Prior research has demonstrated that test-time compute scaling substantially improves performance on mathematical reasoning tasks. Early approaches focused on sampling multiple solution trajectories from LLMs and employing majority-voting-based selection methods (Wang et al., 2022), while more recent breakthroughs such as long reasoning (Jaech et al., 2024; Guo et al., 2025a) scale the length of chain-of-thought reasoning within the model. Contemporary approaches involve generating multiple candidate solutions and selecting the most promising one using a judge model through generative verification (Liu et al., 2025; Toshniwal et al., 2025; Dekoninck et al., 2025), or iteratively refining solutions based on feedback from a judge model (Madaan et al., 2023). In this work, we build upon GenSelect (Toshniwal et al., 2025) and LLM-as-a-Judge (Zheng et al., 2023) methodologies, combining them to achieve superior performance on mathematical reasoning tasks, particularly those requiring formal proofs.

**Mathematical Reasoning Beyond Final Answers.** Recent research focuses on advancing mathematical reasoning beyond simply producing final answers. Datasets such as PRM800K (Lightman et al., 2023) and Process-Bench (Zheng et al., 2025) pioneer the training and evaluation of LLMs on process-level errors, moving past traditional final-answer-based assessment. More recent efforts target proof-based and challenging mathematical problems, including those from the International Mathematical Olympiad (IMO). Mahdavi et al. (2025) evaluate state-of-the-art LLMs on proof-based tasks, revealing common pitfalls such as overgeneralization from limited examples, circular reasoning, and fabricating non-existent theorems. Dekoninck et al. (2025) make a significant contribution to this field by introducing new benchmarks for rigorously evaluating LLMs in the verification and selection of proofs, alongside the first large-scale dataset for this task. They conclude that LLMs are approaching human-level performance in proof evaluation. Ma et al. (2025) introduce a dataset for grading math competition proofs, with a focus on 0-7 point grading, going beyond the binary verification task. Pandit et al. (2025) propose a dataset with a focus on step-level correctness verification of mathematical proofs.

**Reference-Aided Evaluation.** Several recent works show that reference answers, rubrics, or other privileged information can substantially improve LLM-based evaluation. Zhang et al. (2025) improve LLM-as-a-Judge by adapting references to candidate responses, Zhou et al. (2025) show that privileged information can enable expert-level automated evaluation, and Zeng et al. (2024) study LLMs as evaluators for instruction following. These results are complementary to our setting: reference-based checking is often highly effective for final-answer verification and general natural-language evaluation, whereas natural-language proof verification requires judging whether an alternative argument is mathematically valid even when it differs substantially from a canonical proof or rubric.

**Reward Models for Hard-to-Verify Tasks.** In contrast to tasks with easily verifiable answers, proof verification in natural language is an inherently challenging task, even for humans. Recent works have explored training reward models to tackle hard-to-verify tasks. Although no publicly available work specifically explores training LLMs with LLM-based reward models for proof generation, similar approaches have been applied in other domains, such as open-ended writing. The current mainstream approach is to train generative verifiers or use off-the-shelf LLMs to judge the correctness or ranking of model-generated solutions, and use the verifiers to guide the generation of better solutions. Kimi K2 (Team et al., 2025) uses a self-critique-based approach to generate responses and acts as its own critic to judge the response given a predefined rubric for the task. Lu (2025) also use a self-critique approach to rank the responses generated by the underlying LLM for open-ended writing tasks. These rankings are used to produce rewards for training a reward model, which is then used to further finetune the LLM using reinforcement learning.

## 3. Background and Setup

We begin by introducing the two primary paradigms for automated proof verification using large language models: LLM-as-a-Judge and GenSelect. In the LLM-as-a-Judge

paradigm (Whitehouse et al., 2025; Zhang et al., 2024; Shi & Jin, 2025), a language model directly evaluates the correctness of a single proof, typically producing either a binary judgement or a fine-grained score (e.g., on a 7-point scale). To improve reliability, practitioners often sample multiple independent judgements and aggregate them by averaging scores or taking majority votes. We denote the number of sampled judgements per proof as $n_j$. Since this method assigns a numerical score to each proof, it is useful when evaluating individual proofs, as well as for selecting the best proof among multiple candidates by choosing the proof with the highest score. GenSelect (Toshniwal et al., 2025; Dekoninck et al., 2025) takes a comparative approach: rather than evaluating proofs individually, the model compares many proofs in-context, and selects the best one. This method is often used in a pairwise format, and a tournament structure is employed to select the best proof among multiple candidates. A *knockout tournament* operates in single-elimination format, requiring exactly $n_p - 1$ comparisons to select a winner from $n_p$ candidates. Alternatively, a *pairwise tournament* compares all $\binom{n_p}{2}$ pairs and selects the proof winning the most comparisons. This achieves higher accuracy than knockout tournaments but requires quadratically many comparisons. When multiple candidate proofs are available for a single problem, the goal is to select the best proof from among them. We now describe the datasets used to evaluate these methods.

### 3.1. Datasets Details

Evaluating proof verification requires datasets that test different capabilities. We use six complementary datasets, which we name as follows: (1) **VerProofArena:** A collection of proofs from USAMO 2025, IMO 2025, and IMC 2025, with human-labeled judgements from Math-Arena (Balunović et al., 2025). We augment this dataset with recent solutions to IMO 2025 from Gemini (DeepMind, 2025) and OpenAI (OpenAI, 2025a) to improve diversity. This dataset contains 204 proofs with binary correctness labels. The majority of proofs are graded by at least two human graders or are officially accepted solutions. Although the dataset has the highest label reliability, the majority of correct proofs are generated by Gemini. We discuss potential implications of the dataset composition in Section 4. We use this dataset to evaluate proof verification methods. (2) **VerProofBench:** This benchmark includes 429 proofs from recent proof-based competitions, scored on a 7-point rubric by expert human graders from ProofBench (Ma et al., 2025). The task requires judging proof quality on this 7-point scale. In this work, we binarize the scores into correct (score $\geq 6$) and incorrect (score $< 6$) for proof verification judgement. (3) **VerOPC:** A large scale dataset of mathematical proofs with human-labeled correctness from OPC (Dekoninck et al., 2025). This dataset contains 3K training proofs, and 292 test

proofs. (4) **Challenge-19:** We select the 19 most challenging problems from AIME 2024, HMMT 2024, and CMIMC 2025 with integer final answers (called Challenge-19). We retain only problems where GPT-OSS-120B (Agarwal et al., 2025) with high-reasoning mode generates at least one correct answer but maintains a solve rate below 70%. These problems serve dual purposes: evaluating test-time compute methods and assessing proof-level judgement precision on generated solutions. (5) **SelOPC:** The pass@n subset from the OPC (Dekoninck et al., 2025) dataset consisting of 60 problems with 8 proofs per problem generated by the OpenAI o4-mini model. We use this dataset to evaluate selection of the best proof among 8 generated proofs per problem. (6) **SelProofBench:** The best-of-n subset from the ProofBench (Ma et al., 2025) dataset consisting of 29 problems, each with 16 proofs generated by OpenAI o3. We use this dataset to evaluate proof-selection among 16 generated proofs per problem. Of the 29 problems, 14 have at least one correct proof (*i.e.*, score at least 6/7).

## 4. Challenges in Building Reliable Evaluation Sets

We find that the choice of evaluation set is crucial for developing and benchmarking proof judgement models. We identify two primary sources of challenge in building reliable evaluation sets.

**Challenge 1: Human Label Noise.** Human label noise can be significant, making it difficult to obtain large-scale reliable ground truth judgement labels. While VerOPC (Dekoninck et al., 2025) represents a valuable contribution as the first large-scale training set for proof verification with over 3K training examples, the noise in human labels poses challenges for evaluating and training the strongest models. For instance, GPT-OSS-120B achieves $87.67\%$ accuracy on the VerOPC (maj@8), whereas the estimated human labeler accuracy on this set is $90.4\%$ (Dekoninck et al., 2025). This small gap between model and human performance indicates that the VerOPC test set is not suitable for evaluating strong proof graders, as the noise ceiling is too low. Such limitations make the dataset challenging to use for further training strong models such as GPT-OSS-120B via RL. Consequently, in the main experiments, we only use the training set to train a weaker model, Qwen3-30B-A3, as a proof grader.

**Challenge 2: Dataset Imbalances.** Model and problem imbalances in the dataset may lead models to exploit spurious correlations to achieve high accuracy without genuinely understanding the mathematical content of proofs. In such settings, a verifier can appear strong by learning which model styles or problem domains are associated with correct labels. We demonstrate this issue through two case studies using VerOPC (Dekoninck et al., 2025).

*Case Study 1: Simple MLP Classifier.* We train a two-layer Multi-Layer Perceptron (MLP) binary classifier on text embeddings of `(problem, proof)` pairs using a small Qwen3-0.6B (Yang et al., 2025) embedding model. Despite having no explicit mathematical checking capability, this model achieves 75.34% accuracy on the OPC test set, outperforming Claude-Sonnet-4 by more than 5% (see Table 1 in the Appendix for numerical comparisons to all the LLMs).

*Case Study 2: Trivial Heuristic.* We implement a simple rule-based classifier: predict "incorrect" if the problem contains the word "triangle", "correct" if the proof contains "`-----`", and "incorrect" otherwise. The rule is intentionally crude, but still achieves 65.07% accuracy, outperforming Qwen3-8B and GPT-4.1.

Both methods attain high accuracy without any genuine understanding of proof correctness by exploiting two types of dataset imbalances. First, LLMs frequently generate incorrect geometry proofs, making "predict incorrect for all geometry problems" a surprisingly effective strategy. Second, OpenAI reasoning models, which produce more correct proofs than other LLMs, consistently use "`-----`" to separate proof sections, creating an unintended marker for correctness.

These imbalances echo concerns raised by Shi & Jin (2025) in the context of training judge models for final-answer judgement. They show that including entirely correct or incorrect solutions in the RL pipeline from individual models leads to lower performance than balanced sets where each model produces both correct and incorrect answers. While their work addresses training set composition, we reveal that similar imbalances in test sets can introduce substantial evaluation bias, inflating accuracy metrics without reflecting true verification capability.

To mitigate this issue, we use final-answer correctness as an auxiliary evaluation metric alongside proof-level judgement. We generate an equal number of correct-final-answer and incorrect-final-answer solutions for each problem using GPT-OSS-120B and Qwen3-235B-A22-Thinking-2507. The solutions in this dataset are particularly useful because they allow us to monitor the precision of a judge under a balanced final-answer distribution. That is, if a judge predicts a proof to be correct, precision computes how likely it is that the proof also has the correct final answer. Monitoring precision allows us to distinguish between judges that rely primarily on superficial features (e.g., stylistic cues) versus those that genuinely assess the mathematical content. Because of its balanced construction, this final-answer dataset provides a robust benchmark for precision-based evaluation, and any judgement decision based only on model identity, problem type, or both receives no better than random chance.

> **Takeaway 1**: Reliable evaluation requires proofs with (a) high quality ground-truth labels and (b) a balanced problem-difficulty nature and model diversity. Incorporating a balanced final-answer evaluation set can further help distinguish judges that assess genuine mathematical content from those exploiting stylistic features.

## 5. Proof Verification via Single-Proof Judgement

LLM-as-a-Judge is one of the most widely used methods for verification of hard-to-verify tasks (Team et al., 2025). We start by prompt design, and evaluate three prompts for LLM-as-a-Judge: a general-purpose prompt adapted from ProcessBench (Zheng et al., 2025), and two proof-specific prompts from the OPC (Dekoninck et al., 2025) and a Gemini Agent (Huang & Yang, 2025). To standardize their outputs, we only modify the formatting to use XML tags, leaving the prompt content unchanged. We denote these prompts as `General Summary`, `OPC`, and `GIMO`. Figure 1 (right) reports majority@5 results across proof-level and final-answer judgements. In this figure, the "Proofs" columns refer to VerProofArena, while the "Final Answers" columns refer to Challenge-19. The data indicates that prompt choice has a substantial impact on LLM-as-a-Judge performance: `General Summary` achieves high recall but low precision, `GIMO` achieves high precision but low recall, and `OPC` strikes a balance between the two. Hence, we adopt `OPC` as the default prompt in our main experiments.

Given the high variation in the performance of different prompts, we employ RL to fine-tune a language model to enhance its effectiveness as a judge. We fine-tune a Qwen3-30B-A3B-Thinking-2507 model on the OPC training set using Group Relative Policy Optimization (GRPO) (Shao et al., 2024). The optimization utilizes a binary reward function based on the alignment between the model's judgment and the ground truth. We use `OPC RL` to denote models trained with RL using the OPC prompt. In settings such as `GT-Proof→Rubric`, the model is trained with a ground-truth proof as reference information but evaluated with a rubric; a ground-truth proof is a full reference solution, while a rubric is a more compact checklist of expected arguments or evaluation criteria. We include these transfer settings because different proof datasets provide different kinds of reference information. Figure 1 shows the training curve of the RL training with different prompts. Aligned with prior works (Dekoninck et al., 2025), we observe that RL training significantly improves the performance. Interestingly, we find that this performance gain is prompt agnostic, and the high performance variance across prompts is eliminated, as the performance gap is closed between `OPC` and `GIMO` prompts in Figure 1. Although the `OPC` prompt initially exhibits lower performance compared to the

`GIMO` prompt, RL training leads to convergence to similar results. Notably, the VerOPC and VerProofArena datasets differ significantly in their distribution, which influences model performance discrepancy. The VerProofArena (out-of-distribution) dataset is more challenging, with correct proofs being scarce, having lower label noise, and correct proofs being generated by stronger models. In contrast, the VerOPC (in-distribution) dataset has higher correct proofs and easier problems. These dataset differences explain the discrepancies between validation accuracy and the results shown in the table.

> **Takeaway 2**: LLM-as-a-Judge shows high variation across different prompts. However, RL training can significantly (1) improve performance and (2) reduce prompt sensitivity.

While Reinforcement Learning (RL) improves performance on proof judgement (Figure 1, right), it does not yield a significant improvement in precision for final-answer problems. Although precision increased by approximately 2%, this gain remains within the margin of error (standard deviation), as detailed in Table 5 of the Appendix. We hypothesize that the performance gain stems from better model calibration and the ability to identify stylistic errors, such as missing justifications, rather than improved mathematical reasoning. Improvement will likely require larger datasets of complex proofs and extended reinforcement learning training to enhance the model's performance on mathematical content assessment.

> **Takeaway 3**: RL significantly improves proof-level metrics, but the performance gain mainly serves as a calibration process rather than improved mathematical reasoning.

**Adding a rubric does not significantly improve performance.** Recent studies show that having a ground-truth rubric, reference answer, or other privileged information can help LLMs better judge the quality of generated content in general areas (Arora et al., 2025; Zhang et al., 2025; Zeng et al., 2024), as well as some mathematical evaluation settings (Zhou et al., 2025). Assuming access to a ground-truth solution or rubric could potentially help with training better proof judges by collecting pairs of (`problem, ground-truth-proof`) and using them at training time to guide the training of proof verifiers or generators. We evaluate whether including a ground-truth proof or rubric in the prompt during verification improves the model's ability to judge proofs and enables more effective auto-labeling. Overall, adding a rubric does not lead to a substantial performance gain in proof judgement, except for improvement in final-answer precision (see Figure 1 Right). Surprisingly, models trained without a rubric outperform those trained with rubric on the VerProofArena (out-of-distribution), although they underperform on the VerOPC (in-distribution) validation curves shown in Figure 1. Interestingly, our best-performing model is one that was trained without a rubric but was given a rubric at test time, demonstrating particularly high precision. This suggests that the value of reference information is task-dependent: final-answer checking can often reduce to comparing answers, whereas proof verification requires validating potentially different but still correct arguments.

**Ensembling different judges does not outperform the best single judge.** As an alternative scaling strategy, we evaluate ensembles that average correctness scores across different judges. We test this approach on the SelOPC proof selection task and find that ensembling generally fails to surpass the best individual judge. The only exception occurs when one judge is a model trained on the VerOPC dataset with reinforcement learning (see Figure 4 in the Appendix), which we attribute to adaptation to in-distribution data.

**Step-based Judgement does not improve performance.** Recent work has explored step-based judgement, where the model first decomposes a proof into individual steps by inserting step separators, then evaluates each step independently and aggregates the results to form a final judgement. Guo et al. (2025b) report improved performance using such approaches. We implement a step-based judgement model with two prompts: (1) the model decomposes the proof into steps, and (2) the model is reprompted with each step highlighted and asked to judge whether the step is correct. We aggregate the results by marking a proof as correct only if all steps are judged to be correct. Results are shown in Table 3 in the Appendix. We observe higher precision, but significantly lower recall and accuracy. The strict all-steps-correct requirement likely leads to many false negatives that outweigh the precision gains. Exploring better aggregation methods, RL strategies, or step decomposition techniques is a promising direction for future work to improve step-based judgement performance.

We provide ablations on judgement count and base model choice in Appendix C.

> **Takeaway 4**: Whole-proof LLM-as-a-Judge provides the best performance among different proof judgement strategies. Adding a ground-truth proof or rubric to the prompt slightly improves the performance, but the gain is not significant.

## 6. Revisiting Proof Selection Methods

Two of the most prevalent algorithms for proof selection are GenSelect (Toshniwal et al., 2025; Dekoninck et al., 2025) and LLM-as-a-Judge (Whitehouse et al., 2025; Zhang et al., 2024; Shi & Jin, 2025). Here, we first revisit these two methods and run extensive experiments to compare their per-

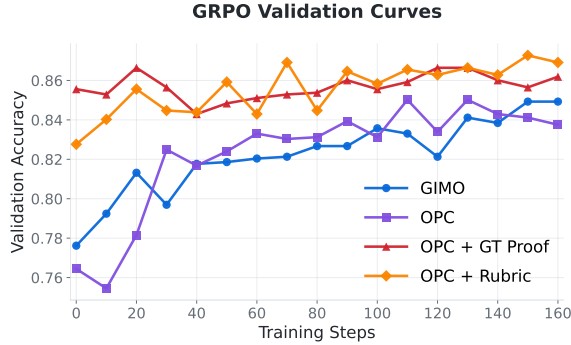

| Prompt | Proofs | | | Final Answers* | |
|---|---|---|---|---|---|
| | Prec | Rec | F1 | Prec | Rec |
| General Summary | 18.28 | 99.89 | 30.91 | 58.21 | 74.86 |
| GIMO | 57.58 | 87.06 | 69.28 | 64.85 | 15.53 |
| OPC | 22.88 | 99.96 | 37.23 | 60.41 | 67.43 |
| OPC (Rubric) | 24.46 | 99.89 | 39.30 | 92.25 | 77.45 |
| OPC RL | 54.03 | 94.57 | 68.75 | 62.59 | 17.18 |
| GIMO RL | 49.66 | 95.81 | 65.40 | 66.29 | 22.65 |
| OPC RL (Rubric) | 39.72 | 98.98 | 56.69 | 98.15 | 50.02 |
| OPC RL (GT-Proof→Rubric) | 39.09 | 98.64 | 55.99 | 98.12 | 49.06 |
| OPC RL (No-Rubric→Rubric) | 59.74 | 95.09 | 73.35 | 99.31 | 21.27 |

*Figure 1. Left:* Training curves for RL training of Qwen3-30B-A3B-Thinking-2507 on the OPC dataset. (1) Different prompts converge to similar performance after RL training. (2) Including the ground-truth proof or rubric in the prompt improves validation performance by 2–3%. (3) Training with both rubric and ground-truth proof achieves comparable performance. *Right:* Majority@5 results for the VerProofArena and Challenge-19 evaluation using Qwen3-30B-A3. See Tables 5 and 6 in the Appendix for full results, additional models, datasets, and prompt ablations. $a \rightarrow b$ denotes trained on $a$ evaluated on $b$. *For final-answer problems, precision and recall are computed based solely on the correctness of the final answer.

formance on existing best-of-n proof selection benchmarks, namely SelProofBench and SelOPC. Dekoninck et al. (2025) report that a pairwise GenSelect tournament outperforms LLM-as-a-Judge on the OPC-PassN benchmark, while Ma et al. (2025) report that LLM-as-a-Judge, using a 7-point grading prompt outperforms the GenSelect tournament. Furthermore, both studies show that 7-point grading prompts outperform binary judgement prompts for LLM-as-a-Judge. To reconcile these conflicts, we conduct experiments on LLM-as-a-Judge and GenSelect using two models: GPT-OSS-120B and Qwen3-235B-A22-Thinking-2507.

First, we scale the number of sampled judgements per proof in LLM-as-a-Judge, as a higher number of judgements generally leads to better estimates of the proof correctness (Shi & Jin, 2025). Indeed, more judgements improve proof selection up to 32 judgements per proof, after which the performance saturates (see Figure 4 in the Appendix). For the Pairwise GenSelect tournament, we use the same setup as Ma et al. (2025), where we perform $\binom{n_p}{2}$ pairwise comparisons, and choose the proof with highest number of wins as the winner. We repeat the experiments for 8 seeds as the results show high variance. The results are depicted in Figure 2. We make the following observations: (1) There is no clear winner between pairwise GenSelect tournament and LLM-as-a-Judge; the best method varies by dataset and model. (2) Contrary to prior findings, 7-point grading does not show a consistent advantage over binary grading, although it outperforms binary grading on 7-point scoring metric. This means that the 7-point grading prompt is better at finding the best incorrect proof, although more research is needed as the 7-point-based results on SelProofBench may not transfer to other datasets such as SelOPC and vice versa. (3) Including a ground-truth proof or rubric in the prompt does not lead to better performance for either method, and

in fact, Qwen3-235B shows worse performance when given a ground-truth proof and rubric in the prompt. Despite these findings, we note that the SelProofBench consists of only 29 problems, therefore, 3.44% accuracy difference is only one problem difference. Our findings highlight the challenges in the significant effort required to build reliable evaluation sets for proof selection, as discussed in Section 3.

> **Takeaway 5**: LLM-as-a-Judge and Pairwise GenSelect Tournament are both strong proof selection methods, and there is no clear winner between the two.

## 7. Scaling Proof Selection

A knockout GenSelect tournament uses pairwise comparisons in a knockout tournament format, requiring exactly $n_p - 1$ comparisons to select the best proof from $n_p$ candidates. While this approach is token-efficient, selecting a single winner has sequential depth at least $\log(n_p)$, limiting parallelization. Furthermore, it is less performant than a pairwise tournament that compares all pairs of proofs (Dekoninck et al., 2025; Ma et al., 2025). In contrast, LLM-as-a-Judge evaluates each proof independently by sampling $n_j$ judgements per proof, requiring $n_j \times n_p$ LLM calls total. This method is highly parallelizable but more computationally expensive due to the larger number of required evaluations.

When scaling proof generation to hundreds of proofs, pairwise tournaments and LLM-as-a-Judge become prohibitively expensive, while a knockout tournament is less expensive. We propose to unify these two approaches into a single framework that strikes a balance between compute intensiveness, efficiency, and accuracy. Given a problem, we first generate $n_p$ candidate proofs by temperature sampling from a language model. Then we run a knockout GenSelect

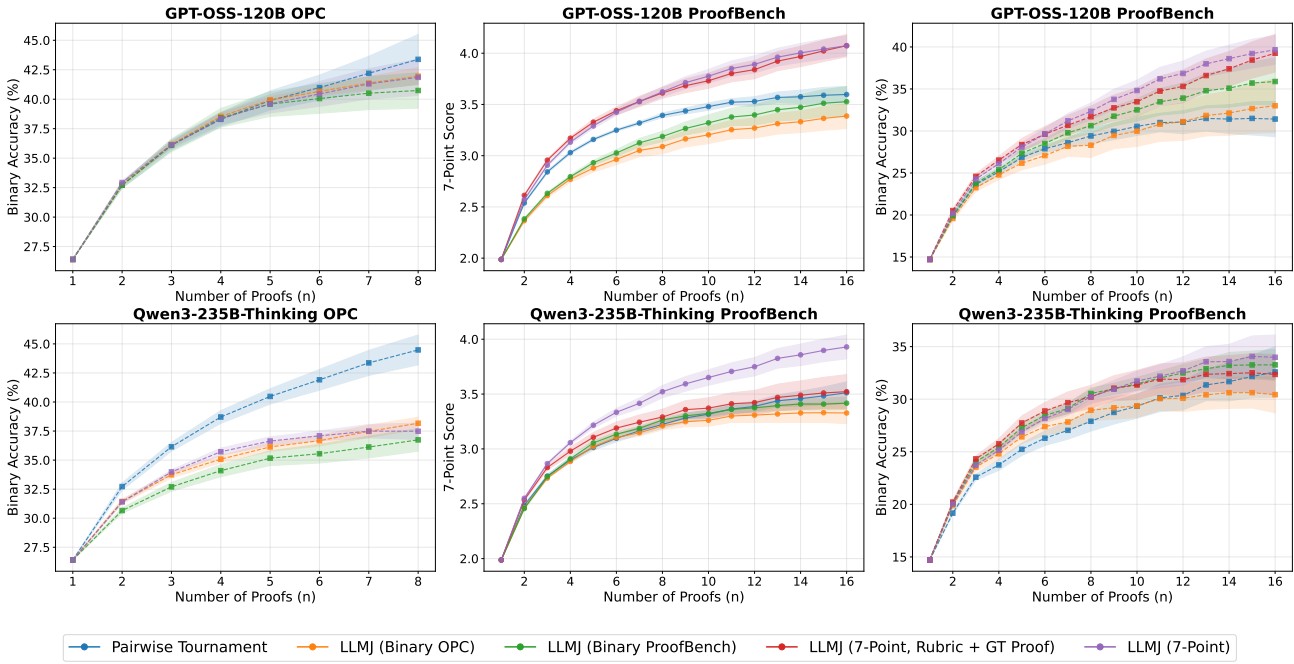

*Figure 2.* Best-of-n proof selection results on SelProofBench and SelOPC using GPT-OSS-120B and Qwen3-235B-A22-Thinking-2507. We compare pairwise GenSelect tournament with LLM-as-a-Judge (averaging 32 judgements per proof) using both binary and 7-point grading prompts. Results are averaged across 8 random seeds. No single method consistently outperforms the others; the best approach varies by dataset and model.

tournament to select the top $n_s$ proofs among the candidates using the same language model. Finally, we run LLM-as-a-Judge to estimate the correctness score of each proof by sampling $n_j$ judgements for each proof, and select the proof with the highest average correctness score as the final proof to solve the problem. This unified framework naturally subsumes both individual methods: setting $n_s = 1$ and $n_j = 0$ reduces to pure knockout GenSelect, while setting $n_s = n_p$ reduces to pure LLM-as-a-Judge. In the hybrid setting, the tournament is stopped once $n_s$ candidates remain, so the sequential depth is roughly $\log(n_p/n_s)$ rather than the full $\log(n_p)$ depth of selecting one winner. This still introduces more latency than a fully pointwise verifier, but our target regime involves thousands of LLM calls per problem, where token efficiency is often the dominant constraint.

**Final Answer Evaluation.** Figure 3 illustrates the proposed method and accuracy results for different configurations on GPT-OSS-120B. We focus our experiments on the final-answer problems since they provide faster evaluation feedback. Our combined approach works on par with, and sometimes outperforms, each method individually. We find that this combined method significantly outperforms majority voting, and consistently matches or outperforms GenSelect (Toshniwal et al., 2025), and is more scalable than LLM-as-a-Judge. Notably, it achieves 100% accuracy on AIME-2025 across all 8 runs using GPT-OSS-120B.

**Proof Evaluation.** As discussed in Section 6, constructing reliable evaluation sets for proof selection is challenging due to the need for a large number of graded proofs to effectively distinguish between different proof selection methods. However, we run preliminary experiments using our test-time scaling method with $n_p = 256, n_s = 16, n_j = 32$ on the USAMO-2025. We focus on problem 2, since it is at the boundary of being solvable by GPT-OSS-120B with high-reasoning mode. We observe that our method selects a correct proof 50% of the time, compared with zero correct proofs from the baseline method (without test-time scaling). We provide a correct example from the test-time scaling method, and an incorrect example from the baseline in Appendix E.

> **Takeaway 6**: Combining a knockout GenSelect tournament with LLM-as-a-Judge for proof selection strikes a balance between compute efficiency and accuracy compared to individual methods.

## 8. Experimental Details

**RL Training.** We train Qwen3-30B-A3B-Thinking-2507 and Qwen3-8B using the DAPO algorithm (Yu et al., 2025) on the VerOPC (Dekoninck et al., 2025) training set. We set aside 277 proofs from the training set as our validation set to monitor the training progress. The training is done on policy, without length penalty, with a batch size of 1024 se-

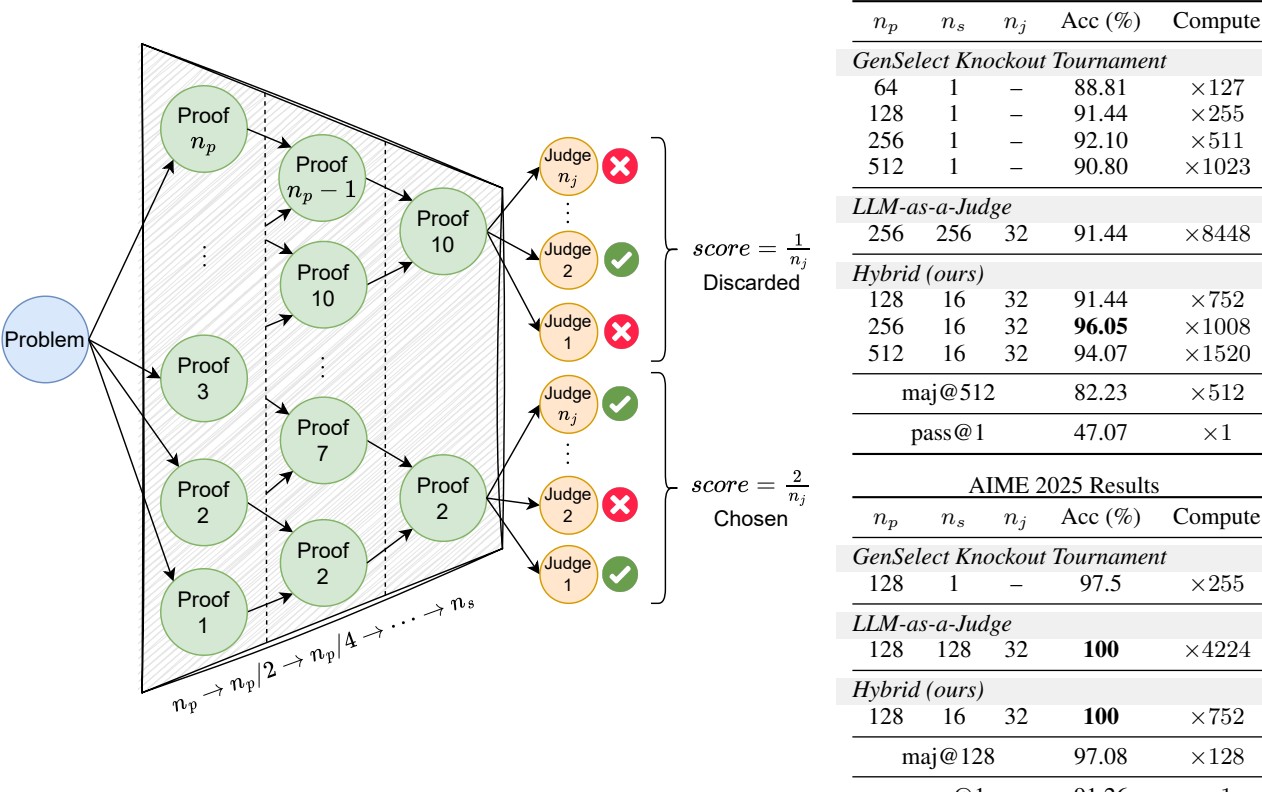

| Challenge-19 Results | | | | |
|---|---|---|---|---|
| $n_p$ | $n_s$ | $n_j$ | Acc (%) | Compute |
| *GenSelect Knockout Tournament* | | | | |
| 64 | 1 | – | 88.81 | ×127 |
| 128 | 1 | – | 91.44 | ×255 |
| 256 | 1 | – | 92.10 | ×511 |
| 512 | 1 | – | 90.80 | ×1023 |
| *LLM-as-a-Judge* | | | | |
| 256 | 256 | 32 | 91.44 | ×8448 |
| *Hybrid (ours)* | | | | |
| 128 | 16 | 32 | 91.44 | ×752 |
| 256 | 16 | 32 | **96.05** | ×1008 |
| 512 | 16 | 32 | 94.07 | ×1520 |
| maj@512 | | | 82.23 | ×512 |
| pass@1 | | | 47.07 | ×1 |
| AIME 2025 Results | | | | |
| $n_p$ | $n_s$ | $n_j$ | Acc (%) | Compute |
| *GenSelect Knockout Tournament* | | | | |
| 128 | 1 | – | 97.5 | ×255 |
| *LLM-as-a-Judge* | | | | |
| 128 | 128 | 32 | **100** | ×4224 |
| *Hybrid (ours)* | | | | |
| 128 | 16 | 32 | **100** | ×752 |
| maj@128 | | | 97.08 | ×128 |
| pass@1 | | | 91.26 | ×1 |

*Figure 3. Left:* Overview of our proposed test-time scaling method to combine GenSelect with LLM-as-a-Judge for proof selection. Given a question, we first generate $n_p$ candidate proofs by sampling from a language model. Then we run a knockout GenSelect tournament using the same language model to select the top $n_s$ proofs. We estimate the correctness score of each proof using LLM-as-a-Judge by sampling $n_j$ judgements for each proof. Finally, we select the proof with the highest average correctness score as the final proof to solve the problem. *Right:* The tables show accuracy results for different configurations of our method on GPT-OSS-120B, averaged across 8 seeds on AIME-2025 and Challenge-19. Compute is reported as the total number of LLM calls per problem. The hybrid combination of LLM-as-a-Judge with GenSelect performs substantially better than GenSelect alone, or majority voting.

quences (64 prompts with 16 rollouts each), and maximum generation length of 16384 on 128 H100 GPUs.

**Dataset Details.** In all datasets, we filter out the proofs that exceed 10K Qwen3 tokens, as they are generally too long or contain many thinking steps, and are all incorrect proofs. We use the rubrics and ground-truth proofs as provided by the datasets, and generate rubrics using LLMs when needed. For the ProofBench datasets, we use the ground-truth proof and rubrics provided by the dataset (Ma et al., 2025). For the proofs taken from MathArena (Dekoninck et al., 2025), we use the provided rubric directly. For the OPC datasets (Dekoninck et al., 2025), we directly use the ground-truth proof provided in the dataset, and generate a rubric using GPT-OSS-120B based on the ground-truth proof when needed. For the final-answer problems, we use the expected final answer as the rubric. This makes verification with rubric a simpler task as the model only needs to compare the two answers and does not need to verify other computational steps or details.

# 9. Discussion: Are LLMs ready to be reliable Mathematical Judges?

While all SOTA LLMs are able to achieve high accuracy in proof judgement (*e.g..*, $> 90\%$ accuracy on ProofBench (Ma et al., 2025)), we observe that this still does not translate to high reliability, especially when scaling test-time compute or judging hard problems. For instance, we attempted to use GPT-5 (high thinking mode) to judge proofs generated by GPT-OSS using various test-time scaling methods in Section 7 on the USAMO 2025 hard problems (problems 5 and 6). For each problem, we generated proofs using six test-time scaling configurations: three pure GenSelect methods, one LLM-as-a-Judge method, and two hybrid methods. Running each configuration with four random seeds produced 24 candidate proofs per problem. We then used GPT-5 as a preliminary filter, aiming to automate the grading process at scale for faster feedback, and sent the GPT-5-positive candidates to expert human graders for final evaluation.

Surprisingly, GPT-5 judged 7 out of the 48 candidates as correct, but expert grading found that all 7 contained critical mathematical errors. Although GPT-5 has $41/48 = 87.2\%$ accuracy on judging these proofs, this still means that it cannot be fully trusted to reliably judge mathematical proofs, especially when the correct proofs are rare. We found that providing a ground-truth solution in the prompt or using majority voting over five judgements did not alleviate this issue. One example of such an incorrect judgement is shown in Appendix E. Notably, this issue persists despite the fact that the generator (GPT-OSS) is significantly weaker than the verifier (GPT-5). Therefore, we advise caution when using large language models as mathematical judges in high-stakes scenarios, especially for complex problems where correct solutions are infrequent, and human verification is necessary.

## 10. Conclusion

In this work, we addressed the critical challenge of verifying and selecting natural-language mathematical proofs using LLMs. We showed that a unified test-time scaling approach that combines GenSelect tournaments with LLM-as-a-Judge evaluation achieves the best performance at scale. We made surprising findings: while reinforcement learning eliminates prompt variation and improves proof-level metrics, it does not improve final-answer precision, suggesting that current approaches may rely more on recognizing stylistic features rather than deep mathematical understanding.

**Future Directions.** An important direction for future work is to focus on problems at the frontier of current LLMs' solving capabilities. By carefully selecting such challenging problems and rigorously labeling them, we aim to create high-quality benchmarks that drive further improvements in proof-judgement and problem-solving models. We also plan to leverage RL to further scale up the performance of LLM-based judges and solvers, as well as explore other scaling methods such as step-level judgement, where intermediate reasoning steps are supervised rather than the final outcome. A particularly important open problem is decoupling stylistic calibration from logical validity in verifier training. Recent concurrent work such as DeepSeekMath-V2 (Shao et al., 2025) explores meta-verification, where a second verifier checks whether the judge's rationale constitutes a mathematically valid justification for its score. This direction is aligned with our finding that binary outcome rewards alone may improve proof-level metrics without improving final-answer precision, suggesting that future reward objectives should explicitly evaluate the logical soundness of the verifier's reasoning.

**Limitations.** While we provide an extensive study over the current state-of-the-art LLMs for proof verification and selection, our experiments and conclusions are limited to the

models and datasets available at the time of writing. It might be possible that future models, trained exclusively on LLM-as-a-Judge or GenSelect tasks, could yield different results. Furthermore, as model capabilities improve, the effectiveness or brittleness of prompts, as well as the effectiveness of rubrics within those prompts, may also change.

## Impact Statement

This paper presents work whose goal is to advance the field of Machine Learning, specifically in the area of mathematical reasoning and proof verification. The development of reliable proof verification systems could benefit mathematics education and research by providing automated feedback on mathematical arguments. There are many potential societal consequences of our work, none which we feel must be specifically highlighted here.

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

## A. Further Experimental Details

**Inference Settings.** We allocate a 100K completion length budget to all models, and use the recommended sampling parameters by the model providers. For GPT-OSS models, we sample with temperature 1.0 and top-p 1.0; for Qwen models, we sample with temperature 0.6, top-p 0.95, and top-k 20. For other models, we use a temperature of 0.7 and top-p 0.95. We use Nemo-Skills (nem, 2025) as our codebase, and SGLang (sgl, 2025) and VLLM (vll, 2025) for efficient LLM inference. All inference workloads are executed on a mix of A100, H100, and GB200 GPUs.

## B. Pitfalls of imbalanced evaluation set

*Table 1.* Benchmarking LLMs as proof graders. All LLM numbers are taken from Dekoninck et al. (2025). We observe that a simple MLP classifier and a simple heuristic can outperform many LLMs.

| Judge | pass@1 |
|---|---|
| Human | 90.4 |
| Gemini-2.5-Pro | 85.4 |
| OPC-R1-8B | 83.8 |
| OpenAI-O4-Mini | 83.8 |
| OpenAI-O3 | 83.1 |
| Gemini-2.5-Flash | 82.7 |
| Qwen3-235B-A22 | 81.8 |
| DeepSeek-R1 | 80.9 |
| **MLP Classifier** | 75.3 |
| Qwen3-30B-A3 | 74.0 |
| DeepSeek-R1-Qwen3-8B | 70.7 |
| Claude-4-Sonnet | 70.6 |
| **Format Heuristic** | 65.07 |
| Qwen3-8B | 64.4 |
| GPT-4.1 | 61.4 |
| Baseline | 53.2 |

## C. Further Ablation Experiments

**GenSelect on judgements improves proof verification performance.** Inspecting the judgements generated by LLM-as-a-Judge, we find that the LLM often finds the mistake in at least one of the sampled judgements. This motivates us to use GenSelect to select the best judgement among multiple judgements generated by the LLM. Since GenSelect shows strong performance in proof selection, we hypothesize that it can also help in selecting better judgements. That is, given a problem, a proof, and $n_j$ judgements, we use GenSelect to select the best judgement among the $n_j$ judgements and use that judgement to evaluate the proof. We run this GenSelect process for $n_j$ seeds and report the majority@5 results in Table 4. We observe improvements in precision and F1 for both GPT-OSS-120 and Qwen3-30B-A3-Thinking, while recall drops slightly. However, when incorporating GenSelect into proof-selection (i.e., selecting the best proof among candidates), we observe no performance improvement. This suggests that GenSelect is particularly beneficial for absolute judgement selection, where choosing a threshold is not possible, but it does not provide gains in the proof-selection setting.

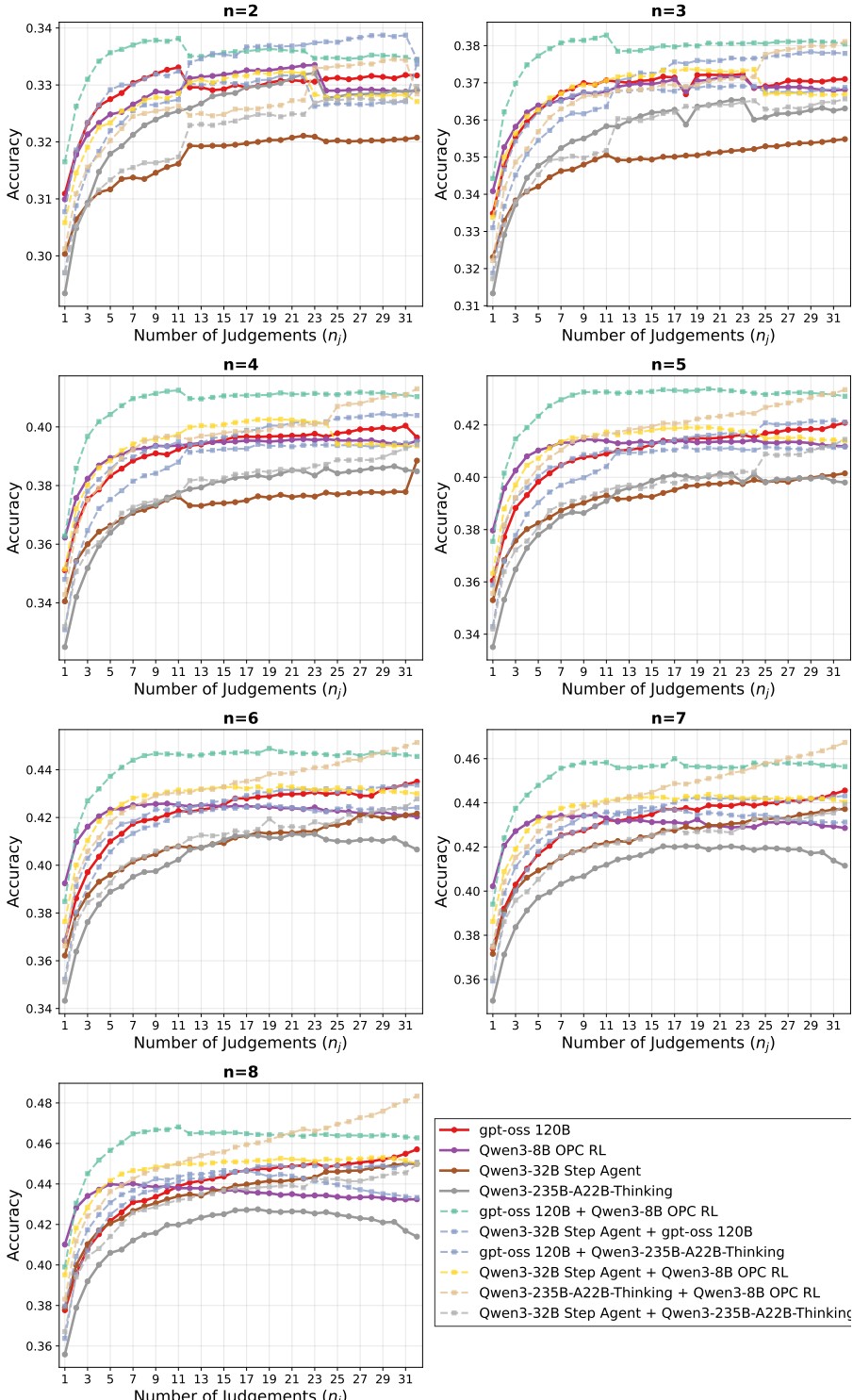

*Figure 4.* LLM-as-a-Judge performance on selecting the best proof among 8 generated proofs from SelOPC. A higher number of judgements leads to better performance, plateauing around 20-30 judgements. Furthermore, ensembling multiple models does not lead to any performance gain, and often the result does not exceed the strongest LLM, except when ensembling an 8B model trained with RL on the training set of OPC. We suspect that the improvement comes from the fact that the 8B model trained with RL is trained on the same distribution as the test set, and therefore is complementary to other LLMs.

*Table 2.* List of challenging final-answer problems from HMMT, AIME, and CMIMC used in the Challenge-19 dataset. All problems have an integer final answer.

| Index | Problem ID |
|---|---|
| 1 | CMIMC 2025 P5 |
| 2 | CMIMC 2025 P7 |
| 3 | CMIMC 2025 P18 |
| 4 | CMIMC 2025 P34 |
| 5 | AIME 2024 P7 |
| 6 | AIME 2024 P14 |
| 7 | AIME 2024 P26 |
| 8 | HMMT Nov 2024 HMIC 1 |
| 9 | HMMT Nov 2024 Guts 19 |
| 10 | HMMT Nov 2024 Theme 7 |
| 11 | HMMT Nov 2024 Theme 9 |
| 12 | HMMT Feb 2024 Guts 28 |
| 13 | HMMT Feb 2024 Guts 29 |
| 14 | HMMT Feb 2024 Combinatorics 1 |
| 15 | HMMT Feb 2024 Combinatorics 4 |
| 16 | HMMT Feb 2024 Combinatorics 9 |
| 17 | HMMT Feb 2024 Combinatorics 10 |
| 18 | HMMT Feb 2024 Algebra 10 |
| 19 | HMMT Feb 2025 Team 4 |

*Table 3.* Comparison of step-based judgment versus vanilla LLM-as-a-Judge on VerProofBench using GPT-OSS-120B. Results are averaged over majority@5 across 32 seeds. Step-based judgment shows higher precision but significantly lower recall compared to vanilla LLM-as-a-Judge.

| Method | Precision | Recall | F1 | Correct Judgements |
|---|---|---|---|---|
| Vanilla LLM-as-a-Judge | $73.62 \pm 1.76$ | $\mathbf{78.51 \pm 2.18}$ | $\mathbf{75.97 \pm 1.60}$ | $\mathbf{89.12 \pm 0.72}$ |
| Step-based Judgment | $\mathbf{89.43 \pm 2.43}$ | $36.36 \pm 1.29$ | $51.69 \pm 1.46$ | $85.11 \pm 0.39$ |

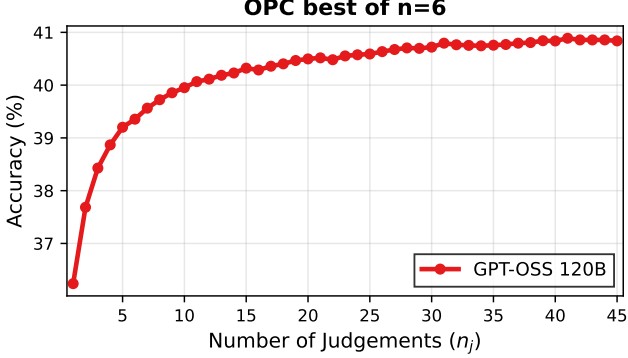

*Figure 5.* GPT-OSS-120B LLM-as-a-Judge performance on selecting the best proof among 6 generated proofs from OPC pass@n subset. We vary the number of judgements from 1 to 45 and observe that the performance saturates at around 32 judgements.

*Table 4.* GPT-OSS-120B and Qwen3-30B-A3-Thinking evaluation. We report precision, recall, and F1 for proof problems, and precision for final-answer problems.

| Model | Judgement | VerProofArena | | | Final-Answer (Challenge-19) |
|---|---|---|---|---|---|
| | | Precision | Recall | F1 | Precision |
| GPT-OSS-120 | LLM-as-a-Judge | 46.28 | **99.17** | 63.10 | 86.18 |
| GPT-OSS-120 | Judge GenSelect | **54.46** | 91.21 | **68.19** | **90.13** |
| Qwen3-30B-A3-Thinking | LLM-as-a-Judge | 22.08 | **99.96** | 37.23 | 60.41 |
| Qwen3-30B-A3-Thinking | Judge GenSelect | **30.65** | 94.49 | **46.29** | **66.26** |

*Table 5.* Majority@5 Results for VerProofArena and Challenge-19 Evaluation sets, averaged across 32 seeds. We ablate various prompts under Prompt $i$ (Ablation).

| Prompt | VerProofArena | | | | Final Answers (Challenge-19) | | |
|---|---|---|---|---|---|---|---|
| | Precision | Recall | F1 | Avg Tokens | Precision | Recall | Avg Tokens |
| **Qwen3-30B-A3B-Thinking-2507** | | | | | | | |
| General Summary | $18.28 \pm 0.28$ | $99.89 \pm 0.45$ | $30.91 \pm 0.41$ | 8587 | $58.21 \pm 1.37$ | $74.86 \pm 2.19$ | 11813 |
| GIMO | $57.58 \pm 2.30$ | $87.06 \pm 2.95$ | $69.28 \pm 2.09$ | 8503 | $64.85 \pm 5.88$ | $15.53 \pm 2.39$ | 10025 |
| GIMO (Ablation 1) | $56.01 \pm 1.95$ | $88.72 \pm 2.34$ | $68.65 \pm 1.87$ | 8664 | $62.04 \pm 4.59$ | $17.14 \pm 1.83$ | 10352 |
| GIMO (Ablation 2) | $54.69 \pm 2.53$ | $88.98 \pm 2.31$ | $67.70 \pm 2.17$ | 8779 | $65.31 \pm 3.97$ | $18.55 \pm 1.74$ | 10716 |
| OPC | $22.88 \pm 0.57$ | $99.96 \pm 0.26$ | $37.23 \pm 0.76$ | 6327 | $60.41 \pm 1.52$ | $67.43 \pm 2.30$ | 8609 |
| OPC (Rubric) | $24.46 \pm 0.51$ | $99.89 \pm 0.45$ | $39.30 \pm 0.67$ | 6046 | $92.25 \pm 1.44$ | $77.45 \pm 2.25$ | 10211 |
| Prompt 1 (Ablation) | $19.43 \pm 0.30$ | $100.00$ | $32.54 \pm 0.42$ | 7276 | $58.23 \pm 1.40$ | $78.39 \pm 1.70$ | 9793 |
| Prompt 2 (Ablation) | $20.68 \pm 0.35$ | $100.00$ | $34.27 \pm 0.49$ | 6856 | $59.21 \pm 1.21$ | $73.08 \pm 1.91$ | 9323 |
| Prompt 3 (Ablation) | $21.99 \pm 0.47$ | $99.36 \pm 0.97$ | $36.01 \pm 0.65$ | 7774 | $61.42 \pm 1.32$ | $76.27 \pm 1.92$ | 10994 |
| Prompt 4 (Ablation) | $23.77 \pm 0.46$ | $99.96 \pm 0.26$ | $38.41 \pm 0.60$ | 6634 | $60.32 \pm 1.90$ | $64.41 \pm 2.73$ | 8995 |
| Prompt 5 (Ablation) | $16.65 \pm 0.21$ | $99.96 \pm 0.26$ | $28.54 \pm 0.32$ | 7757 | $56.64 \pm 1.17$ | $81.63 \pm 1.98$ | 11123 |
| Prompt 5 (Rubric) | $22.27 \pm 0.38$ | $96.53 \pm 1.15$ | $36.18 \pm 0.55$ | 5999 | $98.54 \pm 0.64$ | $94.76 \pm 1.35$ | 11273 |
| Prompt 6 (Rubric) | $27.28 \pm 0.69$ | $99.02 \pm 1.15$ | $42.77 \pm 0.88$ | 5656 | $95.21 \pm 1.68$ | $74.61 \pm 2.77$ | 10182 |
| OPC RL | $54.03 \pm 1.81$ | $94.57 \pm 2.44$ | $68.75 \pm 1.76$ | 5271 | $62.59 \pm 3.59$ | $17.18 \pm 1.50$ | 6070 |
| OPC RL (Rubric) | $39.72 \pm 1.06$ | $98.98 \pm 1.32$ | $56.69 \pm 1.15$ | 5380 | $98.15 \pm 0.68$ | $50.02 \pm 2.60$ | 8238 |
| OPC RL (GT-Proof→Rubric) | $39.09 \pm 1.24$ | $98.64 \pm 1.25$ | $55.99 \pm 1.38$ | 5362 | $98.12 \pm 1.28$ | $49.06 \pm 2.67$ | 8216 |
| OPC RL (No-Rubric→Rubric) | $59.74 \pm 2.53$ | $95.09 \pm 2.33$ | $73.35 \pm 2.27$ | 4997 | $99.31 \pm 1.59$ | $21.27 \pm 1.95$ | 6897 |
| GIMO RL | $49.66 \pm 1.50$ | $95.81 \pm 2.21$ | $65.40 \pm 1.52$ | 6008 | $66.29 \pm 3.47$ | $22.65 \pm 1.87$ | 7041 |
| **GPT-OSS-120B** | | | | | | | |
| General Summary | $41.82 \pm 1.44$ | $95.47 \pm 1.73$ | $58.15 \pm 1.57$ | 17443 | $86.32 \pm 2.53$ | $52.67 \pm 2.31$ | 30709 |
| GIMO | $94.11 \pm 3.26$ | $45.09 \pm 3.07$ | $60.90 \pm 2.96$ | 15599 | $94.60 \pm 12.18$ | $2.00 \pm 1.41$ | 24512 |
| GIMO (Ablation 1) | $96.22 \pm 2.78$ | $54.26 \pm 3.95$ | $69.29 \pm 3.24$ | 12933 | $83.50 \pm 26.90$ | $1.84 \pm 1.09$ | 21280 |
| GIMO (Ablation 2) | $94.82 \pm 3.08$ | $55.13 \pm 4.36$ | $69.62 \pm 3.62$ | 12977 | $85.78 \pm 19.59$ | $2.86 \pm 1.33$ | 22172 |
| OPC | $46.28 \pm 1.24$ | $99.17 \pm 1.08$ | $63.10 \pm 1.22$ | 16807 | $86.18 \pm 2.53$ | $50.29 \pm 1.89$ | 32529 |
| OPC (Rubric) | $44.32 \pm 1.29$ | $99.89 \pm 0.45$ | $61.39 \pm 1.25$ | 10502 | $99.61 \pm 0.79$ | $49.22 \pm 2.70$ | 16848 |
| Prompt 1 (Ablation) | $47.65 \pm 1.40$ | $97.96 \pm 1.12$ | $64.10 \pm 1.38$ | 11267 | $82.23 \pm 2.38$ | $37.84 \pm 2.42$ | 24112 |
| Prompt 2 (Ablation) | $46.99 \pm 0.94$ | $98.04 \pm 1.36$ | $63.52 \pm 0.97$ | 10616 | $81.56 \pm 2.47$ | $33.35 \pm 1.59$ | 21565 |
| Prompt 3 (Ablation) | $48.29 \pm 1.39$ | $97.21 \pm 1.15$ | $64.52 \pm 1.34$ | 10468 | $83.16 \pm 2.07$ | $34.88 \pm 1.98$ | 21268 |
| Prompt 4 (Ablation) | $46.12 \pm 1.00$ | $98.83 \pm 1.30$ | $62.88 \pm 1.07$ | 11388 | $81.75 \pm 1.78$ | $39.06 \pm 2.30$ | 20479 |
| Prompt 5 (Ablation) | $44.55 \pm 1.21$ | $99.58 \pm 0.78$ | $61.56 \pm 1.21$ | 12478 | $85.22 \pm 1.88$ | $49.12 \pm 2.00$ | 24336 |
| Prompt 5 (Rubric) | $47.36 \pm 1.25$ | $89.51 \pm 1.74$ | $61.93 \pm 1.29$ | 4587 | $100.00$ | $67.39 \pm 2.87$ | 13153 |
| Prompt 6 (Rubric) | $46.47 \pm 1.18$ | $98.34 \pm 1.29$ | $63.11 \pm 1.26$ | 9273 | $99.81 \pm 0.57$ | $50.76 \pm 2.33$ | 16418 |
| **GLM-4.5-Air** | | | | | | | |
| General Summary | $20.49 \pm 0.56$ | $94.53 \pm 1.86$ | $33.68 \pm 0.82$ | 16168 | $62.01 \pm 1.92$ | $60.65 \pm 2.82$ | 20373 |
| GIMO | $54.63 \pm 3.16$ | $83.89 \pm 3.50$ | $66.13 \pm 3.04$ | 11006 | $70.35 \pm 6.45$ | $20.75 \pm 2.65$ | 13866 |
| OPC | $20.90 \pm 0.49$ | $95.89 \pm 1.72$ | $34.32 \pm 0.71$ | 13331 | $67.88 \pm 2.14$ | $66.02 \pm 2.74$ | 19184 |

*Table 6.* Majority@5 results for VerProofBench. The setting is similar to Table 5.

| Prompt | VerProofBench | | | |
|---|---|---|---|---|
| | Precision | Recall | F1 | Avg Tokens |
| **Qwen3-30B-A3B-Thinking-2507** | | | | |
| General Summary | 33.40 ± 0.68 | 86.55 ± 1.20 | 48.19 ± 0.78 | 8389 |
| GIMO | 75.76 ± 2.34 | 55.81 ± 1.99 | 64.25 ± 1.83 | 8048 |
| GIMO (Ablation 1) | 77.45 ± 2.58 | 56.81 ± 2.02 | 65.52 ± 1.85 | 8128 |
| GIMO (Ablation 2) | 73.73 ± 2.12 | 58.23 ± 1.76 | 65.05 ± 1.38 | 8231 |
| OPC | 39.16 ± 0.70 | 91.13 ± 1.01 | 54.77 ± 0.75 | 6150 |
| OPC (Rubric) | 60.76 ± 1.47 | 84.74 ± 2.18 | 70.77 ± 1.57 | 4796 |
| Prompt 1 (Ablation) | 36.16 ± 0.67 | 89.96 ± 1.23 | 51.58 ± 0.79 | 7123 |
| Prompt 2 (Ablation) | 38.34 ± 0.78 | 90.51 ± 1.20 | 53.86 ± 0.88 | 6637 |
| Prompt 3 (Ablation) | 39.20 ± 0.82 | 88.15 ± 1.35 | 54.26 ± 0.94 | 7180 |
| Prompt 4 (Ablation) | 43.25 ± 0.99 | 86.98 ± 1.81 | 57.76 ± 1.16 | 6277 |
| Prompt 5 (Ablation) | 32.04 ± 0.51 | 91.26 ± 0.96 | 47.43 ± 0.65 | 7466 |
| Prompt 5 (Rubric) | 59.84 ± 1.42 | 88.02 ± 1.36 | 71.24 ± 1.23 | 4904 |
| Prompt 6 (Rubric) | 63.54 ± 1.67 | 82.26 ± 1.65 | 71.68 ± 1.39 | 4650 |
| OPC RL | 73.66 ± 1.93 | 66.15 ± 2.28 | 69.68 ± 1.81 | 4644 |
| OPC RL (Rubric) | 77.21 ± 1.76 | 75.85 ± 1.62 | 76.51 ± 1.33 | 3926 |
| OPC RL (GT-Proof→Rubric) | 75.79 ± 1.70 | 76.19 ± 1.90 | 75.98 ± 1.56 | 3948 |
| OPC RL (No-Rubric→Rubric) | 86.71 ± 2.48 | 59.00 ± 2.39 | 70.19 ± 2.00 | 3747 |
| **GPT-OSS-120B** | | | | |
| General Summary | 67.66 ± 2.37 | 67.09 ± 1.25 | 67.35 ± 1.41 | 14980 |
| GIMO | 94.95 ± 1.88 | 26.19 ± 1.88 | 41.02 ± 2.33 | 12723 |
| GIMO (Ablation 1) | 95.50 ± 1.35 | 26.15 ± 1.58 | 41.03 ± 1.96 | 12534 |
| GIMO (Ablation 2) | 95.03 ± 1.89 | 26.53 ± 2.30 | 41.43 ± 2.86 | 12628 |
| OPC | 73.62 ± 1.76 | 78.51 ± 2.18 | 75.97 ± 1.60 | 12650 |
| OPC (Rubric) | 80.00 ± 1.55 | 79.91 ± 1.98 | 79.94 ± 1.42 | 6672 |
| Prompt 1 (Ablation) | 76.97 ± 1.66 | 70.98 ± 1.56 | 73.84 ± 1.18 | 11351 |
| Prompt 2 (Ablation) | 77.92 ± 1.85 | 71.02 ± 1.45 | 74.30 ± 1.18 | 9934 |
| Prompt 3 (Ablation) | 78.76 ± 1.96 | 69.57 ± 1.56 | 73.87 ± 1.33 | 9526 |
| Prompt 4 (Ablation) | 76.06 ± 1.83 | 74.91 ± 2.07 | 75.47 ± 1.53 | 10781 |
| Prompt 5 (Ablation) | 74.75 ± 1.67 | 74.15 ± 1.43 | 74.44 ± 1.27 | 12130 |
| Prompt 5 (Rubric) | 81.77 ± 1.76 | 80.06 ± 1.58 | 80.89 ± 1.23 | 4609 |
| Prompt 6 (Rubric) | 81.11 ± 1.84 | 79.72 ± 1.85 | 80.40 ± 1.59 | 5902 |
| **GLM-4.5-Air** | | | | |
| General Summary | 37.86 ± 0.89 | 82.04 ± 1.72 | 51.80 ± 1.05 | 14638 |
| GIMO | 72.90 ± 2.66 | 60.85 ± 3.18 | 66.29 ± 2.60 | 10539 |
| OPC | 42.47 ± 1.02 | 88.89 ± 1.61 | 57.47 ± 1.08 | 12437 |

# D. Prompts

We take our proof generation prompt from the OPC (Dekoninck et al., 2025). To generate proofs for final-answer problems, we add one more sentence to the prompt and ask it to put its final answer within a \boxed{} tag.

---

## OPC Proof Generation Prompt

Your task is to write a proof solution to the following problem. Your proof will be graded by human judges for accuracy, thoroughness, and clarity. When you write your proof, follow these guidelines:

– You are creating a proof, not a proof outline. Each step should be carefully explained and documented. If not properly explained, the judge will assume that you cannot explain it, and therefore decrease your grade.
– You can use general theorems and lemmas, but only if they are well–known. As a rule of thumb: if the result has a name and is famous enough to have a Wikipedia page or something similar to describe it, it is allowed. Any result from papers that would not be taught in high–school or low–level bachelor courses in mathematics should not be used. Any use of such results will immediately give you a zero grade.
– Do not skip computation steps in your proof. Clearly explain what transformations were done and why they are allowed in each step of a calculation.
– You should use correct LaTeX notation to write equations and mathematical symbols. You should encompass these equations in appropriate symbols (”\(” and ”\)” for inline math, ”\[” and ”\]” for block math) to enhance the clarity of your proof. Do not use any unicode characters.
– Your proof should be self–contained.
– If you are not sure about a specific step, or do not know how to prove an intermediate result, clearly state this. It is much preferable to indicate your uncertainty rather than making incorrect statements or claims.

{problem}

---

## OPC Judgement Prompt

You are judging the correctness of an LLM–generated proof for a math problem.

### Input:

Your input will consist of the following components:
– **Problem Statement**: A mathematical problem that the proof is attempting to solve.
– **Proof Solution**: The proof that you need to evaluate. This proof may contain errors, omissions, or unclear steps. The proof was generated by another language model, which was given the following instructions:
<model_prompt>
– You are creating a proof, not a proof outline. Each step should be carefully explained and documented. If not properly explained, the judge will assume that you cannot explain it, and therefore decrease your grade.
– You can use general theorems and lemmas, but only if they are well–known. As a rule of thumb: if the result has a name and is famous enough to have a Wikipedia page or something similar to describe it, it is allowed. Any result from papers that would not be taught in high school or low–level bachelor courses in mathematics should not be used. Any use of such results will immediately give you a zero grade.
– Do not skip computation steps in your proof. Clearly explain what transformations were done and why they are allowed in each step of a calculation.
– You should use correct LaTeX notation to write equations and mathematical symbols. You should encompass these equations in appropriate symbols (”\\(” and ”\\)” for inline math, ”\\[” and ”\\]” for block math) to enhance the clarity of your proof. Do not use any unicode characters.
– Your proof should be self–contained.
– If you are not sure about a specific step, or do not know how to prove an intermediate result, clearly state this. It is much preferable to indicate your uncertainty rather than making incorrect statements or claims.
</model_prompt>

### How the solution should be graded:
A solution should be considered correct even if it would earn 5+/7 points in a standard grading format. Examples of small penalties worth 1 point are if the solution:
– Makes a small computational mistake that can be easily fixed
– Misses an edge case which can be easily proven/disproven
– Skips over a step that follows without much reasoning or manual work

Depending on the severity and the context, you may also not penalise a given error. On the other hand, a solution should be marked as incorrect if:
− It marks a step as trivial, if it is not immediately obvious with little reasoning why this would be the case.
− It omits algebra−heavy computational steps, regardless of whether or not it has outlined the methodology. Skipping shorter computations should be permitted.
− Generalizes over a pattern without rigorously describing the pattern, or without proving any relevant properties.
− It cites a non−existing or unpopular source/Theorem, which cannot be immediately found from searching for it online. Thus, any theorems that can be immediately found and have a Wikipedia article are allowed.

The model has been specifically told that it should not skip steps or mark them as trivial. Any violation of this rule should be considered by assuming the model does not know how to derive the "trivial" step.

### Scoring instructions

If you believe the proof is correct, end your analysis with the following two tags:
<summary>One−paragraph summary explaining why the proof is correct</summary>
<judgement>Judgement: Yes</judgement>.

If you believe the proof is incorrect, end your analysis with the following two tags:
<summary>One−paragraph summary explaining why the proof is incorrect</summary>
<judgement>Judgement: No</judgement>.

### Problem Statement:
{problem}

### Model Solution:
{proof}

---

## General Summary Judgement Prompt

[Instructions]

I will provide a math problem along with a solution. Your task is to review each step of the solution in sequence, analyzing, verifying, and critiquing the reasoning in detail. You need to provide the analyses and the conclusion in the following format :

∗ When you analyze each step, you should use proper verification, recalculation, or reflection to indicate whether it is logically and mathematically valid. Please elaborate on the analysis process carefully.
∗ If an error is detected in any step, you should describe the nature and cause of the error in detail, and suggest how to correct the error or the correct approach. Once a step is found to contain any error, stop further analysis of subsequent steps and provide your judgement.
∗ If no error is detected in any step, you should provide your judgement.

[Format]

After your analysis and conclusion, your response MUST follow this exact format:

For correct solutions, you must end your response with:
<summary>One−paragraph summary explaining why the solution is correct</summary>
<judgement>Judgement: Yes</judgement>

For incorrect solutions, you must end your response with:
<summary>One−paragraph summary explaining why the solution is incorrect</summary>
<judgement>Judgement: No</judgement>

[Problem]

{problem}

[Solution]

{proof}

## GIMO Judgement Prompt

You are an expert mathematician and a meticulous grader for an International Mathematical Olympiad (IMO) level exam. Your primary task is to rigorously verify the provided mathematical solution. A solution is to be judged correct **only if every step is rigorously justified.** A solution that arrives at a correct final answer through flawed reasoning, educated guesses, or with gaps in its arguments must be flagged as incorrect or incomplete.

### Instructions ###

**1. Core Instructions**
* Your sole task is to find and report all issues in the provided solution. You must act as a **verifier**, NOT a solver. **Do NOT attempt to correct the errors or fill the gaps you find.**
* You must perform a **step−by−step** check of the entire solution. This analysis will be presented in a **Detailed Verification Log**, where you justify your assessment of each step: for correct steps, a brief justification suffices; for steps with errors or gaps, you must provide a detailed explanation.

**2. How to Handle Issues in the Solution**
When you identify an issue in a step, you MUST first classify it into one of the following two categories and then follow the specified procedure.

* **a. Critical Error:**
  This is any error that breaks the logical chain of the proof. This includes both **logical fallacies** (e.g., claiming that 'A>B, C>D' implies 'A−C>B−D') and **factual errors** (e.g., a calculation error like '2+3=6').
  * **Procedure:**
    * Explain the specific error and state that it **invalidates the current line of reasoning**.
    * Do NOT check any further steps that rely on this error.
    * You MUST, however, scan the rest of the solution to identify and verify any fully independent parts. For example, if a proof is split into multiple cases, an error in one case does not prevent you from checking the other cases.

* **b. Justification Gap:**
  This is for steps where the conclusion may be correct, but the provided argument is incomplete, hand−wavy, or lacks sufficient rigor.
  * **Procedure:**
    * Explain the gap in the justification.
    * State that you will **assume the step's conclusion is true** for the sake of argument.
    * Then, proceed to verify all subsequent steps to check if the remainder of the argument is sound.

**3. Output Format**
Your response MUST be structured into three XML sections with the tags: <summary>, <detailed_verification>, and <judgement>.

* **a. Summary**
  Wrap this section within <summary>...</summary> tags.
  This section MUST be at the very beginning of your response. It must contain two components:
  * **Final Verdict**: A single, clear sentence declaring the overall validity of the solution. For example: "The solution is correct," "The solution contains a Critical Error and is therefore invalid," or "The solution's approach is viable but contains several Justification Gaps."
  * **List of Findings**: A bulleted list that summarizes **every** issue you discovered. For each finding, you must provide:
    * **Location:** A direct quote of the key phrase or equation where the issue occurs.
    * **Issue:** A brief description of the problem and its classification (**Critical Error** or **Justification Gap**).

* **b. Detailed Verification Log**
  Wrap this section within <detailed_verification>...</detailed_verification> tags.
  Following the summary, provide the full, step−by−step verification log as defined in the Core Instructions. When you refer to a specific part of the solution, **quote the relevant text** to make your reference clear before providing your detailed analysis of that part.

* **c. Judgement**
  This section MUST be at the very end of your response. For correct solutions, you must end your response with EXACTLY:

    <judgement>Judgement: Yes</judgement>
    For incorrect solutions, you must end your response with EXACTLY:
    <judgement>Judgement: No</judgement>

**Example of the Required Summary Format**
*This is a generic example to illustrate the required format. Your findings must be based on the actual solution provided below.*

<summary>
**Final Verdict:** The solution is **invalid** because it contains a Critical Error.

**List of Findings:**
* **Location:** "By interchanging the limit and the integral, we get..."
  * **Issue:** Justification Gap – The solution interchanges a limit and an integral without providing justification, such as proving uniform convergence.
* **Location:** "From $A > B$ and $C > D$, it follows that $A-C > B-D$"
  * **Issue:** Critical Error – This step is a logical fallacy. Subtracting inequalities in this manner is not a valid mathematical operation.
</summary>

=================================================================
### Problem ###

{problem}

=================================================================
### Solution ###

{proof}

=================================================================
### Verification Task Reminder ###

Your task is to act as an IMO grader. Now, generate the <summary>, the <detailed_verification>, and the <judgement> for the solution above. In your log, justify each correct step and explain in detail any errors or justification gaps you find, as specified in the instructions above. End your response with the required XML tags.

---

## Proof GenSelect Prompt

[Instructions]
You are a senior mathematician. You are given a math problem and a list of {num_participants} candidate solutions written by different students.
Your task as a senior mathematician is to carefully analyze the solutions and pick the best solution from the list.

[Format]
Your output must end with the following tags:
<analysis>One–paragraph summary of your analysis of the solutions</analysis>
<best_solution>Index</best_solution>

The best_solution tag MUST ONLY contain an integer between 1 and {num_participants} (inclusive).

---

## ProofBench 7-Point Grading Prompt

You are an **expert math proof grader**. You are judging the
correctness of an LLM–generated proof for a math problem.

### Input

Your input will consist of:

\* \*\*Problem Statement\*\*: A mathematical problem that the proof is attempting to solve.
\* \*\*Proof Solution\*\*: The proof that you need to evaluate. This proof may contain errors, omissions, or unclear steps. The proof was generated by another language model.

### Task

Analyze the proof carefully.

\* Identify logical errors, incorrect steps, or unclear reasoning.
\* Give an \*\*integer\*\* score between 0 and 7 with a brief overall assessment.

### Output Format

Respond with \*\*only\*\* well−formed XML using the structure below. Do not include any extra text or Markdown.

\*\*Requirements:\*\*
− '<score>' must be an integer in [0, 7].
− '<assessment>' must be a \*\*detailed analysis\*\* that explains your reasoning step−by−step and provides a clear \*\*rationale for the score\*\*. Reference specific claims/lines if present.
− '<errors>' must be a list of specific issues (empty if score = 7).

Example output:

<score>0</score>
<assessment>The proof shows a good understanding of the main idea, but has some unclear reasoning and minor mistakes...</assessment>
<errors>
   1. specific error 1,
   2. specific error 2,
   ...
</errors>

### Scoring Guidelines (0−7 scale)

\* \*\*0\*\*: Completely incorrect; proof is irrelevant, nonsensical, or shows no understanding.
\* \*\*1−2\*\*: Very poor; major logical flaws, does not solve the problem, but may contain fragments of relevant reasoning.
\* \*\*3−4\*\*: Partial progress; captures some correct reasoning or key ideas, but has significant logical errors, missing steps, or incomplete arguments that make the proof invalid overall.
\* \*\*5−6\*\*: Largely correct; the proof is overall valid and reaches the correct conclusion. Contains only \*\*minor issues\*\* (e.g., small calculation mistakes, notation slips, or slightly unclear wording) that do not undermine correctness.
\* \*\*7\*\*: Fully correct; the proof is complete, logically sound, and clearly presented with no substantive errors.

________________________________________________________________
\*\*Problem Statement\*\*
{problem}

\*\*Proof Solution\*\*
{proof}

## ProofBench 7-Point Binary Prompt

You are an **expert math proof grader**. You are judging the correctness of an LLM−generated proof for a math problem.

### Input

Your input will consist of:

* **Problem Statement**: A mathematical problem that the proof is attempting to solve.
* **Proof Solution**: The proof that you need to evaluate. This proof may contain errors, omissions, or unclear steps. The proof was generated by another language model.

### Task

Analyze the proof carefully and determine if it is correct or incorrect.

* Identify logical errors, incorrect steps, or unclear reasoning.
* Evaluate the proof using the scoring guidelines below to decide if it should be considered correct (score 6−7) or incorrect (score 0−5).

### Output Format

If you believe the proof is correct (would score 6 or 7), end your analysis with:
<summary>One−paragraph summary explaining why the proof is correct</summary>
<judgement>Judgement: Yes</judgement>

If you believe the proof is incorrect (would score 0−5), end your analysis with:
<summary>One−paragraph summary explaining why the proof is incorrect</summary>
<judgement>Judgement: No</judgement>

### Scoring Guidelines (0−7 scale)

* **0**: Completely incorrect; proof is irrelevant, nonsensical, or shows no understanding.
* **1−2**: Very poor; major logical flaws, does not solve the problem, but may contain fragments of relevant reasoning.
* **3−4**: Partial progress; captures some correct reasoning or key ideas, but has significant logical errors, missing steps, or incomplete arguments that make the proof invalid overall.
* **5−6**: Largely correct; the proof is overall valid and reaches the correct conclusion. Contains only **minor issues** (e.g., small calculation mistakes, notation slips, or slightly unclear wording) that do not undermine correctness.
* **7**: Fully correct; the proof is complete, logically sound, and clearly presented with no substantive errors.

__________________________________________________________________
**Problem Statement**
{problem}

**Proof Solution**
{proof}

## ProofBench 7-Point Grading + Ground Truth + Rubric Prompt

You are an **expert math proof grader**. You are judging the correctness of an LLM−generated proof for a math problem.

### Input

Your input will consist of:

* **Problem Statement**: A mathematical problem that the proof is attempting to solve.
* **Reference Solution**: A correct solution or proof provided for reference. This is **not necessarily the only valid solution**. If the problem requires a final numeric or algebraic answer, this section contains the correct answer, which should be the only accepted final answer (though alternative reasoning paths are valid).
* **Marking Scheme**: A problem−specific grading rubric (0−7 scale) with checkpoints, zero−credit items, and deductions. **Treat this scheme as advisory guidance, not a script.** Use it to anchor scoring, but **do not require** the proof to follow the same order, lemmas, or technique if its reasoning is mathematically sound.
* **Proof Solution**: The proof that you need to evaluate. This proof may contain errors, omissions, or unclear steps. The proof was generated by another language model.

### Task

Analyze the proof carefully.

**Core principles (in order of precedence):**
1) **Mathematical validity** of the proof's reasoning and conclusion.
2) **Problem constraints** (e.g., unique required final value; forbidden tools if stated).
3) **Advisory mapping to the marking scheme** (checkpoints/ deductions), allowing different orders and techniques.
4) **Reference solution** as an anchor for sufficiency, not exclusivity.

**Alternative−approach policy:**
− If the proof uses a different but valid method, **map its steps to equivalent rubric checkpoints** (same logical role) and award points accordingly.
− **Do not penalize** solely for re−ordering steps, using different lemmas, or giving a correct shortcut, **unless** the problem forbids it.
− Apply zero−credit items/deductions **only when the underlying issue actually occurs** in the given proof's approach; **do not auto−penalize** for omitting a rubric step that is unnecessary under the alternative method.
− Avoid double−counting mutually exclusive items; if two items solve the same logical gap, **award the larger only**.
− If the final numeric/algebraic answer is wrong where uniqueness is required, award only partial credit justified by correct intermediate reasoning.

**Rigor and evidence:**
− Award credit for intermediate claims **only if adequately justified** within the proof (not merely asserted).
− If a step is plausible but under−justified, award **conservative partial credit** and note what is missing.

**What to produce:**
− Identify logical errors, incorrect steps, or unclear reasoning.
− Give a **score between 0 and 7** with a **detailed assessment**.
− **Within the assessment text**, show clearly how the score was derived:
    − Which rubric checkpoints (or their **mapped equivalents**)

were earned and the points you awarded.
 – Any zero−credit items or deductions you applied (and why).
 – How these add up to the final integer score in [0−7].

### Output Format

Respond with **only** well−formed XML using the structure below.
Do not include any extra text or Markdown.

**Requirements:**
 – '<score>' must be an integer in [0, 7].
 – '<assessment>' must be a **detailed analysis** that explains
your reasoning step−by−step and provides a clear **rationale for
the score**. Reference specific claims/lines if present. Include
the scoring breakdown **in prose** here (earned checkpoints or
mapped equivalents, deductions, and subtotal −> final score).
 – '<errors>' must be a list of specific issues (empty if score = 7).

Example output:

<score>0</score>
<assessment>The proof shows a good understanding of the main idea,
but has some unclear reasoning and minor mistakes...</assessment>
<errors>
   1. specific error 1,
   2. specific error 2,
   ...
</errors>

————————————————————————————————————————————————————————
**Problem Statement**
{problem}

**Reference Solution**
{ground_truth_proof}

**Marking Scheme**
{rubric}

**Proof Solution**
{proof}

## E. Output Examples

### Incorrect GPT-5-thinking-high Judgement for USAMO2025-P5.

**Problem:** Determine, with proof, all positive integers $k$ such that

$$\frac{1}{n+1}\sum_{i=0}^{n}\binom{n}{i}^{k}$$

is an integer for all positive integers $n$.

- - - - - - - - - - - - - - - - - - - - - - - - - - - - - - - - - - - - - - - - - - -

**GPT-OSS Generated Solution:**

$$A_k(n) = \frac{1}{n+1}\sum_{i=0}^{n}\binom{n}{i}^{k}, \qquad n \geq 1.$$

————————————————————————————————
### 1. Odd exponents do **not** work

Let $k$ be odd and let $p \geq 3$ be a prime. Put $n = p - 1$; then $n + 1 = p$.
For a prime $p$ one has the well-known congruence

$$\binom{p-1}{i} \equiv (-1)^i \pmod{p} \qquad (0 \leq i \leq p - 1) \tag{1}$$

(easily proved from $\binom{p-1}{i} = \dfrac{(p-1)!}{i!(p-1-i)!}$ and Wilson's theorem).

Hence, using (1),

$$\sum_{i=0}^{p-1} \binom{p-1}{i}^k \equiv \sum_{i=0}^{p-1}(-1)^{ik} = \sum_{i=0}^{p-1}(-1)^i \equiv 1 \pmod{p}.$$

The last congruence holds because the number of terms, $p$, is odd. Thus $p$ does **not** divide $\sum_{i=0}^{p-1} \binom{p-1}{i}^k$, so for the admissible $n = p - 1$

$$A_k(n) = \frac{1}{p} \sum_{i=0}^{p-1} \binom{p-1}{i}^k \notin \mathbb{Z}.$$

Consequently no odd integer $k$ satisfies the required property.

### 2. Even exponents always work
Now let $k = 2m$ ($m \geq 1$). For a fixed $n$ denote

$$\mathcal{M}_{n,m} = \left\{ M \in \{0,1\}^{n \times 2m} \;\middle|\; \text{each of the } 2m \text{ columns of } M \text{ contains the same number of 1's} \right\}.$$

If the common column sum equals $i$ then the $i$ positions of the 1's can be chosen independently in each column; therefore

$$|\mathcal{M}_{n,m}| = \sum_{i=0}^{n} \binom{n}{i}^{2m}. \tag{2}$$

#### 2.1 A cyclic action whose orbits have size $n + 1$
Consider a regular $(n + 1)$-gon whose vertices are labelled $0, 1, \ldots, n$ clockwise. Choose one vertex to be *distinguished*; colour it by a special symbol $*$. On every remaining vertex place a colour taken from

$$A = \{0,1\}^{2m} \qquad \text{(the set of binary } 2m\text{-vectors)}.$$

The colouring is required to satisfy

$$\sum_{v \in A \text{ placed}} v = (i, i, \ldots, i) \tag{3}$$

for some integer $i$ ($0 \leq i \leq n$); in other words, the sum of the vectors attached to the $n$ ordinary vertices is a constant vector.
Let $\mathcal{C}_{n,m}$ be the set of all such coloured polygons.
The cyclic group $C_{n+1}$ acts on $\mathcal{C}_{n,m}$ by rotating the polygon one step clockwise. Because the special colour $*$ occurs on exactly one vertex, no non-trivial rotation can fix an element of $\mathcal{C}_{n,m}$; the action is **free**. Hence every orbit of this action contains precisely $n + 1$ elements and therefore

$$|\mathcal{C}_{n,m}| \equiv 0 \pmod{n + 1}. \tag{4}$$

#### 2.2 Bijection with $\mathcal{M}_{n,m}$
Starting from a coloured polygon $P \in \mathcal{C}_{n,m}$ read the colours of the ordinary vertices in clockwise order, beginning with the vertex that follows the distinguished one. The obtained ordered list of $n$ binary vectors is an element of $\mathcal{M}_{n,m}$.
Conversely, given a matrix $M \in \mathcal{M}_{n,m}$, place its rows round a circle in the given order and insert the distinguished vertex after the last row. Because the rows of $M$ have the same column sum, condition (3) is fulfilled, so we obtain a coloured polygon belonging to $\mathcal{C}_{n,m}$.
These two constructions are inverse to each other; consequently

$$|\mathcal{C}_{n,m}| = |\mathcal{M}_{n,m}|. \tag{5}$$

#### 2.3 Divisibility
From (2), (4) and (5) we get

$$n+1 \Bigm| \sum_{i=0}^{n} \binom{n}{i}^{2m}, \qquad \text{hence} \qquad \frac{1}{n+1} \sum_{i=0}^{n} \binom{n}{i}^{2m} \in \mathbb{Z}.$$

Thus the required quotient is an integer for **every** even exponent $k = 2m$.

### 3. Conclusion
The expression $\dfrac{1}{n+1} \displaystyle\sum_{i=0}^{n} \binom{n}{i}^{k}$ is an integer for all positive integers $n$ **iff** $k$ is even.

$$\boxed{k \text{ must be an even positive integer.}}$$

The proof uses only elementary number theory (the congruence $\binom{p-1}{i} \equiv (-1)^i \pmod{p}$) and a simple combinatorial argument (a free cyclic action on coloured polygons). No deep theorems besides these well-known facts are required.

**GPT5-thinking-high Judgement:**
¡summary¿The proof correctly shows that odd k fail by a modular arithmetic argument using the congruence $\binom{p-1}{i} \equiv (-1)^i (mod p)$, and that even k work via a clean combinatorial construction with a free cyclic group action guaranteeing divisibility by n+1. The bijection and counting are clear, and the conclusion matches the ground truth.¡/summary¿
¡judgement¿Judgement: Yes¡/judgement¿

**Ground Truth Judgement:**
The proof is incorrect due to erroneous combinatorial construction.

---

**A correct proof to USAMO2025-P2 provided by our test-time scaling method.**

**Problem:** Let $n$ and $k$ be positive integers with $k < n$. Let $P(x)$ be a polynomial of degree $n$ with real coefficients, nonzero constant term, and no repeated roots. Suppose that for any real numbers $a_0, a_1, \ldots, a_k$ such that the polynomial $a_k x^k + \cdots + a_1 x + a_0$ divides $P(x)$, the product $a_0 a_1 \cdots a_k$ is zero. Prove that $P(x)$ has a nonreal root.

- - - - - - - - - - - - - - - - - - - - - - - - - - - - - - - - - - - - - - - - - - - -

**GPT-OSS Generated Solution:**

**Theorem.** 
$$\text{Let } n, k \in \mathbb{N}, \ k < n.$$
$$\text{Let } P(x) = c \prod_{i=1}^{n} (x - r_i), \ c \neq 0, \ r_i \neq 0, \ r_i \neq r_j \ (i \neq j).$$

Assume that

$$\text{for every real } a_0, \ldots, a_k \text{ with } a_k x^k + \cdots + a_0 \mid P(x) \text{ one has } a_0 a_1 \cdots a_k = 0. \tag{1}$$

Then $P$ possesses a non-real zero.

---

### 1. Reformulation in terms of elementary symmetric sums
For a set $S \subseteq \{1, \ldots, n\}$ put

$$Q_S(x) = \prod_{i \in S} (x - r_i) = \sum_{j=0}^{|S|} (-1)^{|S|-j} e_{|S|-j}(S) \, x^j, \tag{2}$$

where

$$e_m(S) = \sum_{\substack{I \subseteq S \\ |I|=m}} \prod_{i \in I} r_i \qquad (m \geq 0)$$

is the elementary symmetric sum of the numbers $\{r_i \mid i \in S\}$ ($e_0(S) = 1$). The coefficient of $x^j$ in (2) is a non-zero constant multiple of $e_{|S|-j}(S)$. Consequently a divisor of degree $k$ of $P$ has a zero coefficient **iff** for the corresponding $k$-subset $S$ we have

$$e_m(S) = 0 \quad \text{for some } m \in \{1, \ldots, k\}. \tag{3}$$

Thus condition (1) is equivalent to
¿ (**∗**) *Every* $k$-subset $S$ of $\{r_1, \dots, r_n\}$ satisfies ¿ (3).

### 2. A combinatorial lemma

Lemma.

Let $R = \{x_1, \dots, x_{k+1}\} \subset \mathbb{R} \setminus \{0\}$ be distinct. Then there exists a $k$-subset $S \subset R$ with $e_m(S) \neq 0$ $(1 \leq m \leq k)$.  (4)

*Proof.* Suppose that (4) is false; then each of the $k + 1$ subsets

$$S_i := R \setminus \{x_i\} \qquad (i = 1, \dots, k + 1)$$

contains a zero elementary symmetric sum. For each $i$ let

$$j_i = \min\{\, m \mid e_m(S_i) = 0 \,\} \in \{1, \dots, k - 1\} \tag{5}$$

(the case $j_i = k$ cannot occur because $e_k(S_i) = \prod_{j \neq i} x_j \neq 0$).
Because there are only $k - 1$ possible values for the numbers $j_i$ and $k + 1$ indices, the pigeon-hole principle yields distinct $i \neq \ell$ with

$$j_i = j_\ell =: j. \tag{6}$$

Set $T := S_i \cap S_\ell = R \setminus \{x_i, x_\ell\}$ (so $|T| = k - 1$). From (2) we have for any $t \in \mathbb{R}$

$$e_j(T \cup \{t\}) = e_j(T) + t\, e_{j-1}(T). \tag{7}$$

Applying (7) with $t = x_i$ and with $t = x_\ell$ and using (6) gives

$$e_j(T) + x_i\, e_{j-1}(T) = 0, \qquad e_j(T) + x_\ell\, e_{j-1}(T) = 0.$$

Subtracting the two equalities and using $x_i \neq x_\ell$ we obtain

$$e_{j-1}(T) = 0, \qquad e_j(T) = 0. \tag{8}$$

Hence the polynomial

$$Q_T(x) = \prod_{t \in T}(x - t) = x^{k-1} + c_{k-2}x^{k-2} + \cdots + c_0$$

has the two consecutive coefficients $c_{k-j} = (-1)^{j-1}e_{j-1}(T)$ and $c_{k-j-1} = (-1)^j e_j(T)$ equal to zero.

**Claim.** A polynomial whose zeros are distinct real numbers cannot have two consecutive zero coefficients.  (9)

*Proof of the claim.* Let $f(x) = x^m + a_{m-1}x^{m-1} + \cdots + a_0$ have distinct real zeros. If $a_r = a_{r-1} = 0$ for some $1 \leq r \leq m - 1$, then

$$f^{(r)}(0) = r!a_r = 0, \qquad f^{(r-1)}(0) = (r-1)!a_{r-1} = 0.$$

Thus $0$ is a common zero of $f^{(r-1)}$ and $f^{(r)}$. For a polynomial with simple real zeros, each derivative also has simple real zeros and the zeros of $f^{(r)}$ interlace those of $f^{(r-1)}$. Therefore two consecutive derivatives cannot share a zero; otherwise that zero would be a multiple root of $f^{(r-1)}$, contradicting simplicity. Hence the assumption is impossible, proving the claim. ∎
Applying the claim to $Q_T$ contradicts (8). Consequently our assumption that every $k$-subset of $R$ is "bad" is false, and (4) holds. ∎

### 3. From the lemma to the polynomial
Assume, to obtain a contradiction, that **all** zeros of $P$ are real. Let

$$R := \{r_1, \dots, r_n\} \subset \mathbb{R} \setminus \{0\} \qquad (n > k).$$

If $|R| > k + 1$ choose an arbitrary subset $R_0 \subset R$ with $|R_0| = k + 1$. Any $k$-subset of $R_0$ is also a $k$-subset of $R$; hence the hypothesis (∗) for $R$ also holds for $R_0$. Applying the Lemma to $R_0$ we obtain a $k$-subset $S \subset R_0$ for which

$$e_m(S) \neq 0 \qquad (1 \leq m \leq k). \tag{10}$$

For this set $S$ the divisor

$$Q_S(x) = \prod_{r_i \in S} (x - r_i)$$

has, by Vieta's formulas, the coefficients $(-1)^m e_m(S)$ $(m = 0, \ldots, k)$; (10) shows that none of them vanishes. Hence $Q_S$ is a divisor of $P$ of degree $k$ **with all coefficients non-zero**, contradicting the assumption (1).

Thus the assumption that all zeros of $P$ are real is impossible; at least one zero of $P$ is non-real. ∎

---

### 4. Remarks on the tools used
* Vieta's formulas relate coefficients of a monic polynomial to the elementary symmetric sums of its zeros – a standard algebraic fact.
* Rolle's theorem (or the equivalent statement that a polynomial with simple real zeros has simple real zeros for all its derivatives) is a classical result from elementary calculus and was used only in the proof of claim (9).
* The pigeon-hole principle and elementary counting were the only combinatorial tools needed.
All arguments are elementary; no deep results beyond those mentioned are required. ∎

---

## An incorrect proof to USAMO2025-P2 without test-time scaling method.

**Problem:** Let $n$ and $k$ be positive integers with $k < n$. Let $P(x)$ be a polynomial of degree $n$ with real coefficients, nonzero constant term, and no repeated roots. Suppose that for any real numbers $a_0$, $a_1$, …, $a_k$ such that the polynomial $a_k x^k + \cdots + a_1 x + a_0$ divides $P(x)$, the product $a_0 a_1 \cdots a_k$ is zero. Prove that $P(x)$ has a nonreal root.

- - - - - - - - - - - - - - - - - - - - - - - - - - - - - - - - - - - - - - - - - - - -

**GPT-OSS Generated Solution:**
**Proof.** Let

$$n > k \geq 1, \qquad P(x) = c \prod_{i=1}^{n} (x - r_i) \qquad (c \neq 0)$$

be a polynomial of degree $n$ with real coefficients, non-zero constant term, and without repeated roots. Thus every root $r_i$ is a non-zero real number.

Assume, for a contradiction, that **all the roots of $P$ are real**. Consequently the roots $r_1, \ldots, r_n$ are distinct non-zero real numbers.

---

### 1. What we have to find
For a subset $S \subset \{1, \ldots, n\}$ of size $|S| = k$ set

$$Q_S(x) = \prod_{i \in S} (x - r_i) = x^k - e_1(S) x^{k-1} + e_2(S) x^{k-2} - \cdots + (-1)^k e_k(S). \tag{1}$$

Here

$$e_j(S) = \sum_{\{i_1, \ldots, i_j\} \subset S} r_{i_1} \cdots r_{i_j} \qquad (1 \leq j \leq k)$$

is the $j$-th elementary symmetric sum of the numbers $\{r_i \mid i \in S\}$. Because none of the $r_i$ equals 0, the constant term $(-1)^k e_k(S) = (-1)^k \prod_{i \in S} r_i$ is never 0.

Hence a divisor $Q_S$ of degree $k$ has **all its coefficients non-zero** iff

$$e_j(S) \neq 0 \quad \text{for every } j = 1, \ldots, k. \tag{2}$$

If we can exhibit a subset $S$ of size $k$ satisfying (2), then the monic divisor $Q_S$ of $P$ has all coefficients non-zero, so the product of its coefficients is non-zero, contradicting the hypothesis that for every real divisor of degree at most $k$ the product of its coefficients must be 0. Therefore it suffices to prove the existence of such a subset $S$.

---

### 2. The combinatorial Nullstellensatz
We shall use the following well-known result (Alon's *Combinatorial Nullstellensatz*).
¿ **Theorem (Combinatorial Nullstellensatz).** ¿ Let $F$ be a field, $f \in F[x_1, \ldots, x_m]$ a polynomial. ¿ Write ¿

$$> f = \sum_{\alpha \in \mathbb{N}^m} c_\alpha x_1^{\alpha_1} \cdots > x_m^{\alpha_m} > \qquad (c_\alpha \in F). >$$

¿ Suppose that for a particular $\beta = (\beta_1, \ldots, \beta_m)$ we ¿ have $c_\beta \neq 0$. Let $S_i \subset F$ be finite sets with ¿ $|S_i| > \beta_i$ for each $i$. Then there exist ¿ $(s_1, \ldots, s_m) \in S_1 \times \cdots \times S_m$ ¿ such that $f(s_1, \ldots, s_m) \neq 0$.

The theorem is valid over any field; we shall apply it over the field $\mathbb{R}$.

### 3. Two auxiliary polynomials
For indeterminates $x_1, \ldots, x_k$ let

$$E_j(x_1, \ldots, x_k) = \sum_{1 \le i_1 < \cdots < i_j \le k} x_{i_1} \cdots x_{i_j} \qquad (1 \le j \le k)$$

be the elementary symmetric polynomial of degree $j$; thus $e_j(S) = E_j(r_{i_1}, \ldots, r_{i_k})$ when $S = \{i_1, \ldots, i_k\}$.
Define

$$F(x_1, \ldots, x_k) = \prod_{j=1}^{k} E_j(x_1, \ldots, x_k). \tag{3}$$

Next introduce the Vandermonde polynomial

$$V(x_1, \ldots, x_k) = \prod_{1 \le i < j \le k} (x_i - x_j). \tag{4}$$

$V$ is non-zero exactly when the arguments are pairwise distinct and its total degree in each variable equals $k - 1$.
Finally set

$$H(x_1, \ldots, x_k) = F(x_1, \ldots, x_k)\, V(x_1, \ldots, x_k). \tag{5}$$

### 4. A monomial with a non-zero coefficient in $H$
Consider the monomial obtained from $F$ by taking from the factor $E_j$ the term $x_1 x_2 \cdots x_j$ (the product of the first $j$ variables). This contributes the exponent vector

$$(k,\, k-1, \ldots, 1). \tag{6}$$

From the Vandermonde polynomial $V$ choose the term $\prod_{i=1}^{k} x_i^{i-1} = x_1^0 x_2^1 \cdots x_k^{k-1}$, which corresponds to the identity permutation in the usual expansion of the determinant. Adding the exponent vectors (6) and $(0, 1, \ldots, k-1)$ gives

$$(k, k, \ldots, k). \tag{7}$$

Hence the monomial

$$M(x_1, \ldots, x_k) = x_1^k x_2^k \cdots x_k^k$$

appears in $H$ with coefficient $1$ ($\neq 0$) (the product of the coefficients $1$ from the two selected monomials). Consequently the polynomial $H$ satisfies the hypothesis of the Nullstellensatz with

$$\beta_1 = \beta_2 = \cdots = \beta_k = k. \tag{8}$$

### 5. Applying the Nullstellensatz
Let

$$S = \{r_1, r_2, \ldots, r_n\} \subset \mathbb{R}.$$

Because the constant term of $P$ is non-zero, none of the $r_i$ is zero; moreover $n > k$. For each variable $x_i$ we take the same set $S_i = S$. Then $|S_i| = n > k = \beta_i$ for all $i$, so the requirements of the Nullstellensatz are fulfilled.
Hence there exist numbers

$$\alpha_1, \ldots, \alpha_k \in S$$

such that

$$H(\alpha_1, \ldots, \alpha_k) \neq 0. \tag{9}$$

Because the factor $V$ divides $H$, (9) implies $V(\alpha_1, \ldots, \alpha_k) \neq 0$; therefore the $\alpha_i$ are **pairwise distinct**. Consequently the set

$$S_0 = \{\alpha_1, \ldots, \alpha_k\} \subset \{r_1, \ldots, r_n\}$$

has exactly $k$ elements.

From the definition (5) we have

$$H(\alpha_1, \ldots, \alpha_k) = F(\alpha_1, \ldots, \alpha_k)\, V(\alpha_1, \ldots, \alpha_k).$$

Since the second factor is non-zero, we must have $F(\alpha_1, \ldots, \alpha_k) \neq 0$. But $F$ is the product of the elementary symmetric polynomials, hence

$$F(\alpha_1, \ldots, \alpha_k) = \prod_{j=1}^{k} E_j(\alpha_1, \ldots, \alpha_k) = \prod_{j=1}^{k} e_j(S_0).$$

Thus each elementary symmetric sum $e_j(S_0)$ is non-zero; i.e. condition (2) holds for $S_0$.

---

### 6. Constructing a divisor with all non-zero coefficients

Consider the monic polynomial

$$Q(x) = \prod_{r \in S_0} (x - r) = x^k - e_1(S_0)x^{k-1} + e_2(S_0)x^{k-2} - \cdots + (-1)^k e_k(S_0).$$

Because each $e_j(S_0) \neq 0$, every coefficient of $Q$ is non-zero. By construction $Q$ divides $P$ (its roots are a subset of the roots of $P$). Consequently the product of the coefficients of $Q$ is non-zero, contradicting the hypothesis of the problem, which states that *any* real divisor of degree at most $k$ must have a zero coefficient.

---

### 7. Conclusion

The assumption that all roots of $P$ are real leads to a contradiction. Therefore $P$ must possess at least one non-real root.

∎

