# OpenReview forum: "Scaling Generative Verifiers For Natural Language Mathematical Proof Verification And Selection"
_ICML.cc/2026/Conference — ICML 2026 regular_

### Official Review · Reviewer_YTLx · 2026-02-16

**Soundness:** 3
**Presentation:** 3
**Significance:** 3
**Originality:** 2
**Overall Recommendation:** 4
**Confidence:** 4

**Summary:**

This paper studies various test-time scaling approaches including LLM judge for math reasoning tasks such as proof and final answer correctness. The author aggregates several existing datasets together to reduce noise and reproduce a few approaches including knockout tournaments, pairwise comparisons. The authors draw several conclusions such as the sensitivity of prompts for LLM judge, the effect of RL on the sensitivity, etc.

**Compliance With Llm Reviewing Policy:**

Affirmed.

**Final Justification:**

The authors have clarified their difference with related work and have also clarified some of my confusions. Therefore, I decide to raise my rating.

**Key Questions For Authors:**

1. Can the authors clarify some of my confusions and questions regarding Section 5 and 9 from above?
2. The authors seem to make a pretty strong claim about the lack of effectiveness of ground truth proof or rubric for grading math related tasks. Several prior work [1,2,3] which authors did not cite seem to suggest otherwise. Could the authors provide some explanation for this?

[1] Zhang, Qiyuan, et al. "Reviseval: Improving llm-as-a-judge via response-adapted references." arXiv preprint arXiv:2410.05193 (2024).

[2] Zhou, Jin Peng, et al. "Graders should cheat: privileged information enables expert-level automated evaluations." arXiv preprint arXiv:2502.10961 (2025).

[3] Zeng, Zhiyuan, et al. "Evaluating large language models at evaluating instruction following." arXiv preprint arXiv:2310.07641 (2023).

**Limitations:**

yes

**Strengths And Weaknesses:**

**Soundness**:
The paper is somewhat sound. The first half until Section 5 is very solid and the findings are very clear. In Section 5, Table 1 starts to cause some confusion for me. For example, what exactly is OPC RL (GT-Proof -> Rubric)? What is the difference between GT-Proof and Rubric? The authors also mentioned two datasets between Line 261 and Line 269 but the table and figure seem to only show Proofs and Final Answers. Is one dataset corresponding to one of them?

**Presentation**: The submission is mostly clearly written although still with some details missing in my opinion. For example, in Section 9 Line 367, the authors mentioned generating proofs using different methods, reaching 24 candidate proofs. What are these different methods and how are the proofs graded (manual or automatic)? How reliable is the grading process?

**Significance and Originality**: Personally I like to see more work on thorough evaluation of existing methods and new observations. The paper clearly has this in mind. There are good insights but some details could be made clearer.

---

> ### Author Rebuttal · Authors · 2026-03-31
>
> We thank the reviewer for the constructive comments and are glad that they found our evaluation thorough. Below we address the main questions raised in the review.
>
> **Q1:** In Section 5, Table 1 starts to cause some confusion for me. For example, what exactly is OPC RL (GT-Proof -> Rubric)? What is the difference between GT-Proof and Rubric?
>
> OPC RL means that we perform reinforcement learning using the OPC prompt. In the setting labeled GT-Proof -> Rubric, the verifier is trained during reinforcement learning with ground-truth proofs (GT-Proof) as input, but at evaluation time it is given rubrics instead. We include this ablation because the available supervision differs across datasets: some provide ground-truth proofs, while others provide only rubrics. As a result, in certain cases the verifier must generalize across these different forms of reference information at test time.
>
> **Q2:** The authors also mentioned two datasets between Line 261 and Line 269 but the table and figure seem to only show Proofs and Final Answers. Is one dataset corresponding to one of them?
>
> The datasets mentioned in Lines 261–269 refer to the ensembling experiments, whose corresponding figure appears in the appendix. In Figure 1, the Proofs dataset refers to VerProofArena, while the Final Answers dataset refers to Challenge-19, as stated in the figure caption. We also provide a full description of all datasets used in the paper in Section 3.1.
>
> **Q3:** In Section 9 Line 367, the authors mentioned generating proofs using different methods, reaching 24 candidate proofs. What are these different methods and how are the proofs graded (manual or automatic)? How reliable is the grading process?
>
> We generate proofs using six test-time scaling configurations, with four random seeds for each, resulting in 24 candidate proofs in total. These six configurations correspond to three pure GenSelect methods, one LLM-as-a-Judge method, and two hybrid methods, following the setup used in Figure 1 using GPT-OSS-120B. The candidate proofs are first filtered using GPT-5-based grading, and the proofs that pass this stage are then evaluated by expert human graders.
>
> To clarify reliability: the final judgments reported in this analysis are based on expert human evaluation rather than solely on automatic grading. The automatic grading is used only as a preliminary filtering step.
>
> **Q4:** The authors seem to make a pretty strong claim about the lack of effectiveness of ground truth proof or rubric for grading math related tasks. Several prior work [1,2,3] which authors did not cite seem to suggest otherwise. Could the authors provide some explanation for this?
>
> Thank you for pointing out [1-3]; we will add citations to them. We believe these results are compatible with ours because they study different evaluation settings.
>
> Zhang et al. [1] focus on natural language tasks such as machine translation and text summarization, whereas our setting is proof verification, so their conclusions do not transfer directly. Zhou et al. [2] are the most closely related to our work: in their math-reasoning setting, evaluation is based on final-answer problems. This is consistent with our own final-answer experiments, where we also observe strong gains from reference-based checking, with precision approaching 100% in some settings (Figure 1). Our claim is narrower: it concerns proof verification, not final-answer verification. Zeng et al. [3], meanwhile, study instruction-following evaluation rather than mathematical reasoning.
>
> In proof verification, a correct solution need not resemble a canonical proof or fixed rubric. Valid proofs may differ substantially in structure, intermediate claims, and style. For instance, in geometry, an LLM may produce a coordinate-based or case-based argument, whereas the human reference may use a shorter and more intuitive proof. Likewise, in functional equations or existence problems, there may be many distinct valid constructions that do not align with a single rubric. In such cases, the verifier must assess the validity of an alternative argument, rather than simply compare it against a reference.
>
> More broadly, we believe the usefulness of rubrics or ground-truth solutions depends strongly on the evaluation task. These methods can be highly effective in some settings, but should be assessed independently for each task rather than assumed to generalize uniformly across domains.
>
> We thank the reviewer again for these clarification questions. We will incorporate the above explanations in the camera-ready version to further improve clarity. Please let us know if there are any remaining concerns.

---

> > ### Author Rebuttal · Reviewer_YTLx · 2026-04-02
> >
> > Thank you for the rebuttal and I decide to raise my rating.

---

### Official Review · Reviewer_W9S1 · 2026-03-11

**Soundness:** 3
**Presentation:** 3
**Significance:** 2
**Originality:** 2
**Overall Recommendation:** 4
**Confidence:** 2

**Summary:**

This paper presents a systematic empirical study of LLM-based generative verification for natural language mathematical proofs, covering both proof-level judgement (determining correctness) and proof selection (choosing the best among candidates). The authors first highlight two challenges in building reliable evaluation sets, i.e., human label noise and dataset imbalances, demonstrating that a simple MLP classifier and a format-based heuristic can outperform several LLM judges on the VerOPC benchmark by exploiting spurious correlations. They then evaluate multiple LLM-as-a-Judge prompt variants and GenSelect (pairwise tournament) across six datasets, finding that prompt design significantly affects performance but reinforcement learning (GRPO on Qwen3-30B) can reduce this sensitivity while improving proof-level metrics. However, RL does not improve final-answer precision, suggesting current models learn stylistic calibration rather than deeper mathematical reasoning. Finally, the authors propose a hybrid test-time scaling framework that first applies a knockout GenSelect tournament to filter top candidates, then uses LLM-as-a-Judge to score and select the final proof, achieving 100% accuracy on AIME 2025 (30 problems) with GPT-OSS-120B. The paper concludes that no single verification method dominates and that reliable proof verification remains an open challenge.

**Compliance With Llm Reviewing Policy:**

Affirmed.

**Final Justification:**

Since all of my concerns have been addressed and the paper is overall solid, I maintain my score.

**Key Questions For Authors:**

1. Given that the scaling experiments are conducted on 19–30 problems, have you considered reporting confidence intervals or bootstrap significance tests? What is the minimum dataset size at which you would consider the hybrid method's advantage over individual methods to be statistically reliable?

2. For the RL finding (Takeaway 3), can you provide a breakdown of how judgements change after RL training, e.g., how many correct -> incorrect vs. incorrect -> correct flips occur, and whether the flips correlate with proof length, problem difficulty, or specific error types? This would help distinguish the "stylistic calibration" hypothesis from alternative explanations such as insufficient training.

3. The paper does not discuss autoformalization-based verification pipelines (e.g., translating natural language proofs to Lean/Isabelle and checking formally) as an alternative or complementary approach. How do you view the trade-off between scaling generative verifiers and investing in autoformalization, especially for competition-level problems where formal verification is increasingly feasible?

4. The hybrid framework introduces three hyperparameters ($n_p$, $n_s$, $n_j$). In practice, how sensitive is performance to these choices, and is there a principled way to allocate a fixed compute budget across the three stages rather than grid search?

**Limitations:**

Yes

**Strengths And Weaknesses:**

## Strengths

1. Thorough evaluation framework exposing dataset pitfalls. The demonstration that a two-layer MLP on text embeddings (75.3% on VerOPC) and a trivial format heuristic (65.07%) can outperform multiple LLM judges is a valuable and eye-opening finding. This serves as a clear cautionary tale for the community about relying on a single benchmark and highlights the need for more carefully constructed evaluation sets.

2. Comprehensive prompt sensitivity analysis. The paper systematically compares three prompt families (General Summary, OPC, GIMO) across multiple models and datasets, with and without rubrics, ground-truth proofs, and RL fine-tuning. The finding that RL eliminates cross-prompt variance (Figure 1, left) is practically useful for practitioners designing verification pipelines.

3. Honest reporting of negative results. The paper candidly reports that ensembling judges does not outperform the best single judge, that adding rubrics does not significantly help, that step-based judgement hurts recall, and that RL does not improve final-answer precision. These negative findings, while individually unsurprising, are collectively informative for the field.

## Weaknesses

1. Limited technical novelty. The core contribution, combining GenSelect knockout tournaments with LLM-as-a-Judge in a cascade, is a straightforward engineering choice that any practitioner would naturally consider when facing compute-accuracy trade-offs. There is no theoretical motivation, no principled framework for choosing $n_s$ and $n_j$, and no analysis of why or when the combination should be synergistic. The paper reads more as a collection of empirical observations than a coherent methodological contribution.

2. Statistically underpowered scaling experiments. The headline results (Figure 3, right) are evaluated on only 19 problems (Challenge-19) and 30 problems (AIME 2025). On Challenge-19, the 3.44% accuracy gap between the hybrid method (96.05%) and the best GenSelect configuration (92.10%) corresponds to roughly one problem. On AIME 2025, pass@1 is already 91.26%, leaving little room for meaningful differentiation. These sample sizes are far too small to draw reliable conclusions about scaling behavior, yet the paper presents them as key evidence for the proposed method.

3. Shallow analysis of the RL finding. Takeaway 3 that RL improves proof-level metrics but not final-answer precision, suggesting stylistic rather than mathematical learning, is the paper's most interesting finding, but it receives insufficient analysis. The paper offers only speculation without supporting evidence. A more convincing treatment would include qualitative error analysis (what types of judgements change after RL?), probing experiments to distinguish stylistic from mathematical features, or analysis of calibration curves before and after RL.

4. lack of a coherent narrative. The paper's six takeaway boxes mostly convey "no clear winner" messages: GenSelect vs. LLM-as-a-Judge (Takeaway 5), adding rubrics (Takeaway 4), ensembling, binary vs. 7-point grading—all are inconclusive or dataset-dependent. Without a unifying explanation for why these results vary, the paper feels like a technical report cataloguing experiments rather than a research paper advancing understanding. The presentation would benefit from a sharper thesis and fewer but deeper investigations.

---

> ### Author Rebuttal · Authors · 2026-03-31
>
> We thank the reviewer for their positive comments. We are encouraged that they find our analysis to be thorough. Due to character limit, we respond to the main concerns below:
>
> **W1:** There is no theoretical motivation, no principled framework for choosing $n_s$ and $n_j$ , and no analysis of why or when the combination should be synergistic.
>
> We appreciate the reviewer’s concern on the need for theoretical motivations. At the same time, we respectfully note that research on mathematical reasoning in LLMs remains largely empirical, with only limited theoretical understanding available in the current literature. Indeed, **one of our main contributions is a careful re-evaluation of existing methods through running experiments on more than 130 <model, dataset, prompt> tuples**, showing that several conclusions reported in prior work, such as the relative merits of LLM-as-a-judge versus GenSelect, the effects of rubrics, and the impact of prompt variations (Sections 4, 5, and 6), do not consistently hold under rigorous evaluation.
>
> Given this context, we motivate our design choices through systematic experimental ablation studies. In Figure 5, we vary $n_j$ and show that performance saturates around $n_j = 32$, which motivates our choice of $n_j = 32$. In addition, Figure 3 (right) presents ablations over different settings of $n_p$, $n_s$, and $n_j$, showing that the hybrid method consistently achieves the best performance. We agree that a stronger theoretical account of when and why this combination is synergistic would be valuable future work. However, we believe that, at the current stage of the field, a rigorous empirical characterization is both necessary and meaningful.
>
> **W2:** Statistically underpowered scaling experiments. The headline results (Figure 3, right) are evaluated on only 19 problems (Challenge-19) and 30 problems (AIME 2025).
>
> We use AIME because it is a standard and widely adopted benchmark for mathematical reasoning in LLMs. For the other competitions (e.g., HMMT), we selected the hardest publicly available recent problems, since easier recent problems are often already saturated by current models and therefore provide limited discrimination. **The small number of such problems is mainly a limitation of the current benchmark landscape**, not a design choice unique to our paper.
>
> To reduce randomness, **all reported results are averaged over 8 random seeds**. Larger-scale evaluation would of course be desirable, but test-time scaling at the extreme budgets studied here is very expensive. For the most resource-intensive method in Table 3, assuming an equivalent API cost of 0.36 per 1M tokens, about 20K output tokens per LLM call, and 8,448 calls per problem, the cost is approximately \$60.82 per problem. **Evaluating a single method on a single LLM for 8 seeds therefore costs about \$23,841 in equivalent API terms**. This makes substantially broader evaluation difficult in practice.
>
> **W3:** Shallow analysis of the RL finding. Takeaway 3 that RL improves proof-level metrics but not final-answer precision, suggesting stylistic rather than mathematical learning, is the paper's most interesting finding, but it receives insufficient analysis. The paper offers only speculation without supporting evidence.
>
> We agree deeper analysis would strengthen the paper, but Takeaway 3 is not purely speculative. The paper already provides two supporting analyses: (1) a balanced final-answer  benchmark designed to separate superficial/stylistic cues from genuine mathematical assessment, on which RL improves proof-level judgment but not final-answer precision; and (2) dataset-imbalance case studies showing that non-mathematical cues can drive strong judge performance, with a small embedding MLP reaching 75.34% accuracy on OPC and a trivial heuristic reaching 65.07%, outperforming several LLM judges. Together, these results support our interpretation that RL mainly improves sensitivity to surface/procedural cues rather than deeper mathematical reasoning. A fuller manual error analysis or calibration study would be valuable, but is costly at our scale and requires substantial expert inspection.
>
> **W4:** lack of a coherent narrative. The paper's six takeaway boxes mostly convey "no clear winner" messages.
>
> Our paper is organized around a single question: how should mathematical verification with LLMs be done when test-time compute is scaled? To answer this, in addition to the negative results, we highlight three positive findings: (1) LLM-as-a-Judge has likely been underestimated in prior work because it was not scaled sufficiently, and with enough sampled judgments it becomes genuinely competitive with GenSelect; (2) RL is useful because it largely removes prompt sensitivity and stabilizes judge behavior; and (3) a hybrid pipeline that uses GenSelect for efficient coarse filtering and LLM-as-a-Judge for final rescoring, which improves performance and gives the best accuracy-compute trade-off at scale.

---

> > ### Author Rebuttal · Reviewer_W9S1 · 2026-04-03
> >
> > Thank you for your response. I have no further questions, and I would like to keep my positive rating.

---

### Official Review · Reviewer_8qmq · 2026-03-13

**Soundness:** 3
**Presentation:** 3
**Significance:** 3
**Originality:** 3
**Overall Recommendation:** 4
**Confidence:** 3

**Summary:**

The authors evaluate two primary paradigms for proof assessment: LLM-as-a-Judge and GenSelect (comparative selection among multiple proofs). By scaling evaluation to millions of tokens across six diverse datasets, they expose critical failures in current benchmarks, such as human label noise and exploitable dataset imbalances where simple heuristics can outperform state-of-the-art models. To address these issues, they propose a hybrid framework that unifies token-efficient knockout tournaments with parallel LLM judgements. This combined approach achieves superior efficiency and accuracy, notably reaching 100% accuracy on AIME 2025 using GPT-OSS-120B

**Compliance With Llm Reviewing Policy:**

Affirmed.

**Key Questions For Authors:**

Please refer to the weakness part.

An interesting finding is that RL improves stylistic calibration but fails to improve genuine mathematical precision. Do you have any idea of proposing a specific training objective to decouple style from logic?

**Limitations:**

Yes

**Strengths And Weaknesses:**

On my side, this paper is well written. I enjoy reading different takeaways.

Here are some strengths and interesting findings:

Rigorous Evaluation suite: The paper exposes significant "shortcuts" in existing datasets, showing that trivial rule-based classifiers (e.g., checking for specific symbols) can achieve high accuracy (65%+) without genuine mathematical understanding.

Effective Hybrid Scaling: The proposed unified framework balances compute and accuracy by using knockout tournaments to narrow down candidates before applying more expensive parallel judging

Here are some weakness (on my perspective)

(Minor point) The proposed "Knockout + Parallel Judge" design has a sequential depth of at least $log(n_p)$ for the tournament phase. While token-efficient, this introduces a latency bottleneck that might be unacceptable for real-time applications compared to a fully parallelizable pointwise verifier.

---

> ### Author Rebuttal · Authors · 2026-03-31
>
> We thank the reviewer for their positive comments. We are encouraged that the reviewer find our paper well-written, our evaluations to be rigorous, and that they found the resulting takeaways interesting. Below, we answer the main questions raised by the reviewer:
>
> **W1:** The proposed "Knockout + Parallel Judge" design has a sequential depth of at least $log(n_p)$ for the tournament phase. While token-efficient, this introduces a latency bottleneck that might be unacceptable for real-time applications compared to a fully parallelizable pointwise verifier.
>
> In our method, the sequential depth is actually $\log(n_p) - \log(n_s)$, since tournament filtering is performed only until the candidate set is reduced to $n_s$ solutions, rather than to a single solution. This is strictly smaller than the depth of a pure Knockout tournament, which is $\log(n_p)$. More importantly, our setting is a high-compute regime in which each problem involves thousands of LLM calls. In this context, token efficiency is a more important consideration than latency in real-time applications.
>
> **Q1:** An interesting finding is that RL improves stylistic calibration but fails to improve genuine mathematical precision. Do you have any idea of proposing a specific training objective to decouple style from logic?
>
> We think this is a great question and a very interesting direction for future work. One promising approach is to augment the verifier’s reward with some form of logic verification in addition to the binary correctness reward. For example, the very recent concurrent work DeepSeekMath-V2 [1] proposes using a meta-verifier to check whether the verifier’s rationale constitutes a mathematically valid justification for the predicted score. Their conclusions are consistent with our findings: RL based only on binary outcome correctness can lead to reward hacking, while adding a meta-verifier helps mitigate this issue. We will add a discussion of this point in the camera-ready version.
>
> [1] Shao et al., DeepSeekMath-V2: Towards Self-Verifiable Mathematical Reasoning

---

> > ### Author Rebuttal · Reviewer_8qmq · 2026-04-02
> >
> > Thank you for your rebuttal. I decide to keep the score positive.

---

### Decision · Program_Chairs · 2026-04-30

**Decision:**

Accept (regular)

**Comment:**

The paper studies how different proof verification methods scale, with the goal of going beyond final-answer verification to improve the quality of the mathematical reasoning and proofs.

Reviewers appreciated the rigorous evaluation done in the paper and were impressed by the empirical results. It seems that the scaling approach that the authors find in the paper is convincingly effective. Some reviewers initially expressed concerns regarding the impact and significance of the results, mentioning limited technical novelty and viewing some of the results as engineering choices that do not provide deep insights on the behavior of RL. However, after discussion with the authors it seems that most of the concerns were resolved, and all reviewers are supportive of the paper being accepted.

In light of this, I recommend that the paper is accepted.